# Ovarian Tumor Biomarkers: Correlation Between Tumor Type and Marker Expression, and Their Role in Guiding Therapeutic Strategies

**DOI:** 10.3390/ijms262311702

**Published:** 2025-12-03

**Authors:** Wiktor Gawełczyk, Julia Soczyńska, Adrian Muzyka, Julia Rydzek, Krzysztof Majcherczyk, Mateusz Żołyniak, Sławomir Woźniak

**Affiliations:** 1Division of Anatomy, Department of Human Morphology and Embryology, Student Scientific Society Anatomia-Klinika Nauka, Wroclaw Medical University, 50-367 Wroclaw, Poland; julia.niznik@student.umw.edu.pl (J.S.); adrian.muzyka@student.umw.edu.pl (A.M.); julia.rydzek@student.umw.edu.pl (J.R.); krzysztof.majcherczyk@student.umw.edu.pl (K.M.); mateusz.zolyniak@student.umw.edu.pl (M.Ż.); 2Division of Anatomy, Department of Human Morphology and Embryology, Wroclaw Medical University, 50-367 Wroclaw, Poland; slawomir.wozniak@umw.edu.pl

**Keywords:** biomarkers, ovarian tumors, sensitivity, specificity, classification

## Abstract

Ovarian tumors constitute a complex and heterogeneous group of neoplasms, encompassing both benign and highly malignant lesions. Accurate diagnosis and classification of ovarian tumor types are crucial for the personalization of therapeutic strategies and have a significant impact on patient prognosis. This review presents the current state of knowledge regarding both classical and novel biomarkers, with particular emphasis on their diagnostic, predictive, and prognostic value. Traditional markers, such as CA-125 and human epididymis protein 4 (HE4), remain central to clinical diagnostics; however, their limitations highlight the need for more sensitive and specific approaches. Emerging biomarkers, including microRNAs (miRNA), circulating tumor DNA (ctDNA), and advanced panels integrating transcriptomic, proteomic, and genomic data, offer the potential for earlier detection, improved disease monitoring, and assessment of treatment response. Despite these advances, major challenges persist, particularly those associated with the heterogeneity of ovarian tumors, the high costs of testing, lack of standardization, and unequal access to diagnostic methods.

## 1. Introduction

The ovary is an organ where physiological and pathological changes frequently occur [1]. It is estimated that 14–18% of postmenopausal women, as well as 7% of reproductive-age women without symptoms present with benign ovarian cysts [2]. In addition to benign tumors, the classification includes borderline ovarian tumors and malignant tumors [3]. Ovarian tumors can also be discussed in the context of pediatric cases. The majority of ovarian neoplasms are benign [4,5]. According to reports, benign tumors may be associated with an increased likelihood of malignant transformation—ovarian cancer [6], which exhibits the highest mortality among gynecological cancers worldwide [7]. According to GLOBOCAN data, there were 324,398 newly diagnosed cases of ovarian cancer and 206,839 deaths attributed to the disease in 2022 [8]. It ranks eighth in terms of incidence among women [9]. The prevalence of specific ovarian cancer subtypes varies by region. For instance, Eastern Europe shows the highest incidence of serous and mucinous ovarian cancers, North Africa demonstrates a predominance of endometrioid carcinoma, and East Asia exhibits a higher frequency of clear cell carcinoma [10]. Ren et al., analyzing ovarian cancer in women aged 45 years and older in their systematic analysis, report a decline in global disease burden between 1990 and 2021 and estimate stabilization of rates over the next 25 years [11]. Early-stage diagnosis is currently possible in approximately 20–25% of patients; however, extensive research efforts are ongoing worldwide to improve early detection rates [12,13]. Malignant tumors progress rapidly and often cause systemic symptoms, in contrast to benign tumors, where symptoms are often subtle and may manifest as abdominal distension or a palpable mass [12]. Reports indicate that the majority of ovarian tumors are diagnosed during routine gynecological examinations or abdominal ultrasonography [14]. Detection of a mass necessitates initiation of a diagnostic pathway. Determining the nature of the lesion—benign or malignant—is critical [15]. Diagnostic differentiation involves the analysis of macroscopic, microscopic, and immunohistochemical properties and can often be challenging. Dundr et al., in their meta-analyses, describe difficulties in distinguishing primary mucinous ovarian tumors from metastases due to overlapping characteristics [16]. Regarding ovarian cancer, Pangaribuan et al., based on analytical diagnostic test study with a cross-sectional approach and using secondary data recorded from medical records, report CA-125 as a significant diagnostic tool with an accuracy of 94.5% (cut-off value 36.5 IU/mL). They emphasize the necessity of concurrent clinical assessment using serum markers and histopathological examination based on histological ones to establish a definitive diagnosis [17]. Some studies suggest combining this with another serum agent—HE4 (cut-off = 70 pmol/L [18]) and preoperative inflammatory markers for differential diagnosis of ovarian cancer from benign lesions [19]. Immunohistochemistry, in turn, allows histological differentiation, for example, in epithelial tumors, typically involving histological markers such as WT1, p53, napsin A, and PR, although other markers are also used. Results are classified into subtypes and evaluated using predictive biomarker tests [20]. In identifying malignant ovarian lesions, certain hormones, proteins, and enzymes (inhibin, chorionic gonadotropin, Alpha-fetoprotein (AFP), lactate dehydrogenase) are also useful [21]. Broadly defined molecular biomarkers are used not only for diagnostics but also for monitoring disease progression and developing targeted therapies. The literature indicates ongoing advances in understanding molecular compositions and emphasizes the need to develop therapeutic variants, highlighting the importance of identifying biomarkers with high sensitivity and specificity and their appropriate selection for clinical purposes [20,22]. Biomarkers are also described in terms of predicting response to immunotherapy, supporting the principles of personalized medicine [23].

The aim of this study is to analyze the existing literature regarding correlations between ovarian tumor types and biomarker profiles and their application in clinical practice. We aim to identify the limitations of biomarker use, considering costs, test availability, and regional disparities. Normanno et al. recently published a survey on biomarker testing access, initiated by the International Quality Network for Pathology, the European Cancer Patient Coalition, and the European Federation of Pharmaceutical Industries and Associations. The study examined both the accessibility and quality of biomarker testing across Europe. The authors reported that access to precision medicine is higher in countries that have public national reimbursement systems. In contrast, limited diagnostic laboratory infrastructure, poor organization and insufficient public reimbursement reduce access to individual biomarker tests in many European countries. In nations with restricted public reimbursement, pharmaceutical companies and patients’ out-of-pocket payments were the main sources of funding for such testing [24]. The pathway from biomarker discovery to clinical implementation requires multiple levels of evidence. The ASCO Tumor Markers Guidelines Committee established five Levels of Evidence (LOE) for determining clinical utility. However, further identification, discovery, and correlational studies are needed to integrate and validate future prognostic biomarkers that can serve as guidelines in ovarian cancer therapy [25]. We attempt to systematize the evidence on classical and emerging markers, considering diagnostic, prognostic, predictive, and monitoring functions. We examine the role of these indicators in clinical practice and identify gaps, thereby outlining potential directions for future research.

## 2. Classification of Ovarian Tumors and Associated Biomarkers

Ovarian tumors are divided into groups based on criteria such as histology, behavior and molecular features. Regarding histological classification, we differentiate three main subtypes: epithelial, germ cell and sex cord–stromal tumors. Epithelial tumors are the most common subtype, which is further divided into five main categories—high-grade serous (70%), endometrioid (10%), clear cell (10%), mucinous (3%) and low-grade serous (<5%) carcinomas [26,27,28,29].

### 2.1. Epithelial Carcinomas

High-grade serous carcinomas (HGSOCs) are aggressive, often presenting at advanced stages. The five-year survival rate for this subtype is, in general, less than 50%, with some of the sources reporting rates as low as 29% for advanced-stage and around 43% for all serous carcinomas combined. Most cases are diagnosed at stage III or stage IV, where the five-year survival drops to 26–46% [30,31,32]. The median overall survival for women with HGSOCs is reported to be approximately 4 years, though it varies widely, depending on factors such as genetic factors, treatment response and stage at diagnosis [33]. HGSOC accounts for 70–80% of ovarian cancer deaths, making it the most lethal gynaecologic malignancy [34,35]. Classic serum markers for HGSOC include CA-125 and HE4. CA-125 is the most established serum marker for this type of malignancy. However, CA-125 alone has limited sensitivity for early-stage disease and can be elevated in benign conditions (e.g., endometriosis, menstruation). Research estimates sensitivity and specificity to be approximately 63–83% and 71–83%, respectively, for CA-125 alone. Another more specific marker is HE4, especially for distinguishing malignant from benign ovarian masses. While the sensitivity of HE4 alone remains around 64–75%, the sensitivity reaches up to approximately 90–97%. The best results are reached when both mentioned markers are used in combination, with reported sensitivity and specificity up to 80–90% and 70–96%, respectively, and AUC reaching 0.96, outperforming either marker alone [36,37,38,39]. Nearly all HGSOCs show abnormal p53 immunostaining, reflecting underlying *TP53* mutations. An observational study by Cole A.J et al. found that 94% (68/72) of cases had these mutations with high or absent p53 expression. Abnormal p53 immunostaining—either diffuse strong nuclear positivity (≥70% nuclei) or complete absence—is typical of HGSOCs, while intermediate expression is rare and most likely indicates wild-type *TP53* [40]. Interestingly, high levels of total Δ133p53 isoform have been observed with improved overall survival in HGSOCs, regardless of chemotherapy response [41,42]. This might be due to the isoform’s ability to suppress the actions of the mutant p53, thereby mitigating its tumor-promoting effects. Studies report Δ133p53 as a marker for improved overall and recurrence-free survival in patients with p53-mutant HGSOCs, with a significant reduction in risk of recurrence and death with hazard ratios for recurrence-free and overall survival: 0.57 and 0.37, respectively [41,43]. No significant association was found between the expression of p53β or p53γ isoforms and survival outcomes in HGSOCs.

Other notable immunohistochemical markers include WT1, PAX8, keratins, Ki67 and p16. WT1 and PAX8 present with the highest sensitivity for HGSOC (often >90%), as in most of the cases, these markers are being expressed. However, the limitation is their specificity, since WT1 and PAX8 are positive in other serous tumors, as well as some non-gynecologic malignancies [44].

Endometrioid ovarian cancer (EOC) typically affects women around 50 years old and is frequently diagnosed at an early stage. It is often low-to-intermediate grade and is strongly associated with endometriosis, as well as Lynch syndrome, though less commonly [45,46]. Multiple large cohort studies report five-year overall survival rates for EOC of about 80–80.6% and a survival rate of around 68.4% for 10 years [47,48]. Survival is the highest for stage I/II. Stage III/IV survival rates are lower, though generally better than other epithelial subtypes. A Canadian 10-year follow-up study estimates the hazard ratio for death to be 0.41 (without adjusting for stage), with serous cancer as a reference [47]. Typically used markers for diagnosis and monitoring of EOC are CA-125 and HE4. Research focused specifically on EOC is limited, though the same as in HGSOC, serum HE4 is generally a better tool than CA-125. Studies show that those markers combined make up for sensitivity being around 82–90% and specificity 91–92% [49]. To increase the sensitivity, Smac protein levels can be added, achieving a sensitivity of 98.33%; however, the specificity of the triple combination drops to 75%, which is lower than HE4 alone [50].

Clear cell ovarian cancer (CCOC) is a distinct and relatively rare subtype of epithelial ovarian cancer. It frequently arises from endometriotic ovarian cysts and is more common in younger women. A large population-based study from 1988 to 2001 showed that the median age at diagnosis for CCOC was 55 years, with a newer Chinese cohort from years 2010–2020 placing the median age at 49.2 years [51,52]. As well as EOC, CCOC is frequently associated with endometriosis, which is considered to be a direct precursor in about 50% of cases [53,54]. The five-year survival rate for CCOC ranges from 60% to 61% in large retrospective and population-based studies, which is slightly lower than for non-clear cell epithelial ovarian cancers (e.g., serous or endometrioid subtypes) [55,56]. CCOCs are defined by a unique set of molecular and protein biomarkers. A panel combining three markers: IMP3, Napsin A, and HNF-1β, achieves high diagnostic accuracy (AUC = 0.954), with both sensitivity and specificity above 80% for distinguishing CCOC from other ovarian cancer subtypes and benign conditions [57]. IMP3 (IGF2BP3) is an mRNA-binding protein that regulates mRNA stability and translation, promoting the proliferation, migration and invasion of cancer cells. Its overexpression is associated with poorer prognosis and resistance to chemotherapy in ovarian cancer, especially in serous, clear cell and mucinous types, and may also be a useful biomarker [58]. Napsin A is an aspartyl protease that is a highly specific marker of clear cell ovarian cancer and allows it to be distinguished from other types of ovarian cancer [59]. HNF-1β (transcription factor) is strongly overexpressed in clear cell ovarian cancer, where it regulates glucose metabolism, promotes the Warburg effect and resistance to apoptosis, and its presence is diagnostically and potentially therapeutically useful [60]. Another highly expressed molecule in CCOC is Cystathionine Gamma-Lyase, which supports tumor survival under stress and is linked to hypoxia-inducible factor 1-alpha expression [61]. Large-scale proteomic profiling of CCOC tissue identified IFITM1 as a highly robust prognostic marker. IFITM1 (interferon-induced transmembrane protein 1) is an immune response-related protein whose epigenetically regulated overexpression promotes migration, invasion, and metastasis in ovarian cancer, and methylation of the IFITM1 promoter may be a biomarker of progression [62]. High IFITM1 expression is significantly correlated with recurrence-free survival and overall survival in CCOC patients, validated across multiple patient cohorts using both mass spectrometry and immunohistochemistry. IFITM1 was one of 15 proteins found to be significantly correlated with recurrence-free survival in CCOC [63].

Mucinous ovarian carcinoma (MOC) is a rather rare and unique subtype of epithelial ovarian cancer. It presents typically as a large, unilateral, multiloculated cystic mass, often exceeding 10 cm in diameter, with smooth external surfaces and variable mucin content in cyst loculi [64,65]. The five-year survival rate for MOC varies significantly by stage and tumor grade at diagnosis, but overall, five-year disease-specific survival is approximately 84–85% for all stages combined [66,67]. MOC has a distinct biomarker profile. A strong expression of MUC1 is observed in primary MOC and is helpful in differentiating from metastatic mucinous tumors. MUC5AC is a mucin, a secretory glycoprotein whose expression is characteristic of mucinous and endometrioid ovarian tumors, and its presence is associated with a lack of lymph node metastasis and a more favorable prognosis [68]. MUC5AC expression decreases as the tumor invasiveness increases, and is higher in benign and borderline tumors, rather than in carcinomas. Another frequently positive biomarker in primary MOC is PAX8, which is used for distinguishing ovarian origin from gastrointestinal metastases. High PAX8 expression is also associated with better prognosis [69]. CK7 is also typically positive, while CK20 and CDX2 are variably expressed. Most notably, a combination of CK7+/SATB2− is highly specific and sensitive for primary MOC, being able to be differentiated with colorectal metastases. The CK7+/SATB2− profile demonstrates a sensitivity of 78% and specificity of up to 99% for identifying primary ovarian mucinous neoplasms. When used together, CK7 and SATB2 achieve an overall diagnostic accuracy of 95.3% in distinguishing primary ovarian mucinous tumors from colorectal and appendiceal metastases, outperforming traditional panels such as CK7/CK20/CDX2 [70,71,72]. Other valuable markers include TFF1, 2 and 3. These proteins are significantly elevated in MOC and may serve as histotype-specific markers [73]. MOC has frequent alterations in *HER2*, *KRAS*, and *TP53* genes. The most mutated gene is *KRAS*, with reported mutation rates ranging from 44% to 68% across large cohorts and populations. These mutations happen early and are considered a part of tumorigenesis [74,75]. *HER2* amplification is observed in approximately 14–33%, with most meta-analyses reporting between 14 and 19%. *HER2* amplification and *KRAS* mutations are often, but not always, mutually exclusive [76,77]. *KRAS* is a small GTPase that cycles between inactive (GDP-bound) and active (GTP-bound) states. Oncogenic mutations in *KRAS* (commonly at codons 12, 13, or 61) lock it in the active, GTP-bound form, leading to persistent downstream signaling. The signal transduction cascade starts with active *KRAS* recruiting and activating RAF kinases (e.g., BRAF, RAF-1). Then, RAF phosphorylates MEK1/2, which is followed by the phosphorylation of ERK1/2. Activated ERK1/2 translocates to the nucleus, regulating transcription factors and gene expressions that drive cell proliferation, survival and differentiation. In summary, constitutive MAPK pathway activation leads to uncontrolled cell growth, resistance to apoptosis and increased tumor aggressiveness in ovarian cancer [78,79,80]. *TP53* mutations are less frequent in early-stage or low-grade MOC but become more common in advanced or high-grade cases, with reported rates from 27% to 75% [81,82].

Low-grade serous carcinoma (LGSC) is the rarest subtype of epithelial ovarian cancer, accounting for less than 5% serous ovarian carcinomas. LGSC typically affects younger women with a median age of 43–55 years [83,84]. Most cases arise from serous borderline tumors, progressing in a stepwise fashion [85]. LGSC is associated with better long-term survival than HGSC, but recurrences are common and often indolent. Large cohort studies report five-year relative survival rates for LGSC, ranging from 61% to 92%, depending on stage at diagnosis and treatment era. A Dutch nationwide study of 855 patients found five-year relative survival rates of 85–92% for stage II disease, with lower rates for more advanced stages [86,87]. Key molecular biomarkers in LGSC are MAPK pathway mutations. The most characteristic molecular feature is activating mutations in genes which control the pathway: *KRAS* (20–47%), *BRAF* (6–10%), and *NRAS* (11%) [88,89]. Less commonly, mutations in *USP9X*, *NF1*, and gene fusions involving *BRAF*, *FGFR2*, or *NF1* are observed [90]. The estrogen receptor is almost universally expressed in LGSC, while progesterone receptor expression is variable (27–90%) [91,92,93]. An obvious need is to differentiate LGSC cases from HGSC cases. Most reliable biomarkers utilized for such differentiation are *TP53*, p16, Ki-67, and MAPK pathway mutations. A highly sensitive and specific combination for LGSC is the combination of wild-type *TP53* and patchy p16 expression, while abnormal *TP53* and block p16 expression indicate HGSC [94]. Figure 1 illustrates the constitutive MAPK pathway activation.

### 2.2. Biomarker Class Performance by Epithelial Carcinoma Subtypes

Although numerous biomarkers have been proposed for epithelial ovarian cancer, their performance is highly context-dependent. Recent evidence indicates that the sensitivity, specificity, and clinical applicability of a given biomarker class—whether serum-based, genetic, or proteomic—are strongly influenced by tumor histotype. A systematic overview of biomarker utility within individual ovarian carcinoma subtypes is, therefore, critical to bridge the gap between molecular research and clinical implementation. Classical Serum Markers (CA-125, HE4), including HGSOC, are widely used, though they have limited sensitivity/specificity, especially for early-stage disease. Multimarker panels (e.g., CA-125, HE4, E-cadherin, IL-6) significantly improve early detection (AUC up to 0.96) and outperform single markers. For CCOC and MOC, CA-125 and HE4 are less sensitive; these subtypes often present with normal or only mildly elevated levels, limiting diagnostic utility. In cases of EOC, combining CA-125 and HE4 enhances risk stratification and early detection, but performance varies by histotype and stage. Finally, in LGSC cases, classical markers are less reliable; it often shows CA-125 elevations, though it has no diagnostic value in distinguishing between subtypes [95,96,97,98]. In terms of immunohistochemical panels, their utility in HGSOC is high, as IHC for WT1, p53, and multimarker panels (e.g., KIAA1324, PLCB1) improve histotype discrimination and prognostic stratification. For CCOC, markers such as Napsin A, HNF-1β, and ARID1A are highly specific for diagnosis and have significant prognostic value. In MOC cases, CK7, CK20, and CDX2 help distinguish primary from metastatic tumors, and in cases of EOC/LGSC, IHC panels can support accurate subtyping and guide therapy selection [98,99,100]. Genetic and transcriptomic markers are fundamental to HGSOC, in which *TP53* mutations are nearly universal. *BRCA1/2* mutations are informative in the context of prognosis and PARP inhibitor therapy. Moreover, transcriptomic signatures (e.g., miR-375, miR-34a-5p) are useful in the prediction of surgical outcomes and recurrence. In CCOC, *ARID1A* and *PIK3CA* mutations are common occurrences. In *MOCs*, *KRAS* mutations are frequent, same as for LGSC, where *KRAS* and *BRAF* mutations are characteristic, with potential for targeted approaches [98,99,101,102] Circulating biomarkers, such as ctDNA and miRNAs, have found their utility, particularly in HGSOC. ctDNA and circulating miRNAs (e.g., miR-1246, miR-375) offer high specificity for diagnosis, early detection, and real-time monitoring of treatment response and recurrence—often outperforming CA-125 in predictive value. Regarding other subtypes, data are emerging; circulating markers show promise but require further validation for subtype-specific use [103,104]. Proteomic panels (e.g., EEF1G, MSLN, BCAM, TAGLN2) and metabolomic signatures (e.g., glycoprotein inflammation markers) achieve high diagnostic accuracy for HGSOC (AUC > 0.93). For CCOC, MOC, and LGSC, proteomic and metabolomic approaches are under investigation; early data suggest potential for improved subtype discrimination and risk prediction [105,106]. A comparative analysis of biomarker class performance has been listed in Table 1 below.

### 2.3. Germ Cell Tumors

Germ cell tumors account for approximately 15–30% of all ovarian tumors, but only 2–5% of ovarian malignancies, with most being benign (primarily mature cystic teratomas). In patients under 20 years old, up to 60% of ovarian tumors are of germ cell origin, and the likelihood of malignancy increases with younger age [107,108,109]. The main classifications of subtypes include teratomas (81–95%), dysgerminoma (7–8%), yolk sac tumor (YST) (2–3%), choriocarcinoma (<1%) and mixed ovarian granulosa cell tumors (3%) [108,110].

Undoubtedly the most frequent group of germ cell tumors are teratomas, occurring across a wide age range, but most frequently in women of reproductive age. The most prevalent type are mature cystic teratomas, which are usually benign, accounting for up to 70% of benign ovarian masses in reproductive-age women. These tumors are typically cystic and contain well-differentiated tissues such as skin, hair, teeth, fat, and sometimes neural tissue. The cyst is often filled with sebaceous material and may have a characteristic “dermoid plug” or Rokitansky nodule [111,112]. A more rare, malignant variant is immature teratomas, which are more common in younger patients. These tumors contain immature or embryonic tissues, often neuroectodermal, and are usually larger and more solid than mature teratomas [113]. The rarest type is monodermal teratomas, composed predominantly of a single tissue type, such as thyroid tissue (struma ovarii) or carcinoid tumor [114]. Benign mature teratomas have an excellent prognosis, with five-year survival rates approaching 100% after surgical removal, as malignant transformation is extremely rare. In the case of immature teratomas, overall five-year disease-specific survival is estimated to be around 91–97% for all stages combined, proved by different multivariate analyses and basic research [115,116,117]. A biomarker that us frequently expressed in immature teratomas is SALL4—a nuclear transcription factor. SALL4 is a master regulator of embryonic stem cell pluripotency, forming a core transcriptional network with OCT4 and NANOG. It directly binds to and activates the enhancers/promoters of these genes, preventing differentiation and maintaining the undifferentiated state [118,119]. SALL4 recruits epigenetic modifiers such as DNA methyltransferases (DNMTs) and histone deacetylases (HDAC1/2), leading to the methylation of CpG islands and deacetylation of histones. This epigenetic silencing represses differentiation genes and maintains the stem cell-like chromatin landscape. SALL4 also regulates histone demethylases (e.g., KDM5B, KDM6A/B), influencing histone marks (H3K4me3, H3K27me3), which are critical for gene expression programs that sustain pluripotency [120,121]. SALL4 associates with the NuRD co-repressor complex to repress tumor-suppressor genes (e.g., Foxl1, Dusp4), which would otherwise inhibit proliferation and promote differentiation. This repression is crucial for the long-term maintenance of undifferentiated, stem-like cells [122]. Additionally, SALL4 activates oncogenic signaling pathways such as Wnt/β-catenin, PI3K/AKT, and Notch, which are known to promote proliferation, survival, and maintenance of stemness [123]. Its presence is associated with higher tumor grades. SALL4 shows high sensitivity in detecting immature teratoma components, with strong nuclear staining in most cases. In a large ovarian germ cell tumor (OGCT) series basic research study, SALL4 was positive in 11 of 15 immature teratomas, indicating a sensitivity of about 73% for this subtype [124]. SALL4 is generally negative in mature teratomas, except for focal staining in primitive neuroepithelial or glandular areas; therefore, SALL4 is not a sensitive marker for mature teratomas, but can highlight immature foci if present. SALL4 is highly specific for germ cell tumors, with minimal staining in non-germ cell ovarian tumors. In a study of 3215 tumor metanalyses, SALL4 was consistently expressed in germ cell tumors but only rarely in non-germ cell tumors, supporting its high specificity [119,123].

Ovarian dysgerminoma is the most common malignant germ cell tumor of the ovary, primarily affecting patients under 30 years old, with a median age of around 22–24 years. Dysgerminoma is more frequent in those with gonadal dysgenesis and can be associated with conditions like Turner syndrome [125]. It typically appears as a large, solid, multilobulated ovarian mass with well-defined borders [126]. Hemorrhage, necrosis, and cystic changes can occur. Bilateral involvement is seen in 10–15% of cases. Most large studies report five-year overall survival rates for dysgerminoma between 94% and 98% across all stages and age groups. Survival decreases with age, with patients ≥50 years having lower five-year survival (approximately 69%) [127,128,129]. Nearly all patients with ovarian dysgerminoma present with markedly elevated serum LDH levels prior to therapy, often several times above the upper limit of normal. This elevation is much more pronounced than in other ovarian tumors, where LDH is less frequently or only mildly elevated. Dysgerminomas show characteristically increased levels of the fast LDH isoenzymes (LDH-1 and LDH-2), which can help distinguish them from other ovarian malignancies that typically elevate the slow isoenzymes. While LDH is not exclusive to dysgerminoma and can be elevated in other malignancies or benign conditions, the degree of elevation and isoenzyme pattern are highly suggestive when combined with clinical and imaging findings [130,131,132]. OCT4 (*POU5F1*) shows strong, diffuse nuclear staining in nearly all dysgerminomas, including metastatic and gonadoblastoma-associated cases. It is highly sensitive and relatively specific, with over 90% of tumor cells positive in most cases. OCT4 is rarely positive in non-dysgerminomatous tumors, making it a central marker for diagnosis [133]. Multiple metanalysis studies demonstrate that OCT4 immunostaining is positive in nearly all cases of ovarian dysgerminoma. In a study of 33 dysgerminomas (including metastatic and gonadoblastoma-associated cases), 100% showed strong, diffuse nuclear OCT4 staining, with more than 90% positive tumor cells in each case. Other series confirm that all or nearly all dysgerminomas are OCT4-positive. OCT4 is highly specific for dysgerminoma among ovarian tumors. In another research study, all 111 non-dysgerminomatous ovarian tumors were negative for OCT4, except for focal weak staining in a minority of clear cell adenocarcinomas (4/14 cases). Other research supports that OCT4 is negative in mature teratomas, YST, Sertoli–Leydig cell tumors, granulosa cell tumors (GCT), and most other ovarian neoplasms. Placental alkaline phosphatase (PLAP) is commonly expressed in dysgerminoma, though less specific than OCT4 [134,135]. PLAP shows very high sensitivity for dysgerminoma. In a retrospective study of 31 ovarian dysgerminoma cases, PLAP was positive in 100% (31/31) by immunohistochemistry [136]. Other case report studies in germ cell tumors (including dysgerminoma and seminoma) report sensitivity rates between 94% and 100% [137,138]. Finally, SALL4 is also expressed in the nuclei of dysgerminoma cells and, as mentioned above, confirms the germ cell origin of the neoplasm [135]. SALL4 in ovarian cancer most often co-occurs with other tumor markers, especially in germ cell and serous tumors, and this co-expression has significant prognostic and diagnostic value. In some cases, particularly in non-germ cell tumors, SALL4 may be present on its own, which may also have diagnostic value [119,123,124,139].

SALL4 is a tumor marker that most often co-occurs with other markers in ovarian cancer, although in some cases, it may be present on its own. In serous ovarian cancer (SOC), SALL4 has been shown to be frequently co-expressed with the ALDH1A1 marker, and high co-expression of these two proteins is significantly associated with advanced disease stage, metastasis and poorer prognosis. High levels of SALL4/ALDH1A1 co-expression are an independent prognostic factor for poor progression-free survival and disease-specific survival, making this co-expression a potential biomarker for ovarian cancer progression [139]. In OGCTs, such as dysgerminoma, mixed germ cell tumors and immature teratoma, SALL4 is strongly expressed and often co-occurs with markers such as OCT3/4, and its presence correlates with a higher degree of malignancy and advancement and shorter progression-free survival. SALL4 positivity is also associated with high levels of AFP and LDH, and OCT3/4 expression further predicts the risk of metastasis [123]. SALL4 is a highly sensitive and specific marker for OGCT and is particularly useful in differentiating yolk sac tumors from clear cell carcinoma, where other markers, such as AFP or glypican-3, show lower sensitivity and specificity. In germ cell tumors, SALL4 is strongly positive in over 90% of cells, whereas in clear cell carcinoma, SALL4 expression is very rare and limited to individual cases [124]. A comprehensive immunohistochemical study has shown that SALL4 is consistently expressed in all germ cell tumors, whereas in non-germ cell tumors, such as serous ovarian cancer, SALL4 expression occurs in approximately 20% of cases, usually in poorly differentiated forms. In these cases, SALL4 may be present without other markers of pluripotency, such as OCT4 or NANOG, which may be diagnostically helpful [119].

YSTs most frequently affect adolescents or young adults (mean age of approximately 21 years), with rare cases in women over 50. It presents as a rapidly enlarging pelvic or abdominal mass, often with pain or distension [140]. Most large studies report overall five-year survival rates for ovarian YST as between 81% and 94%. In advanced stages, survival drops to 25–71%, depending on stage and residual disease after surgery [141,142,143]. AFP is the most established and clinically useful biomarker for ovarian YST. It plays a central role in diagnosis, treatment monitoring, and assessing prognosis. Nearly all ovarian YSTs show markedly elevated serum AFP levels, while it is rarely positive in other ovarian tumors [144,145]. AFP production in YSTs is due to the presence of yolk sac endoderm, mirroring its physiological synthesis during early embryogenesis. AFP activates pathways such as PI3K/AKT, cAMP-PKA-c-fos/c-jun, and WNT/β-catenin, leading to increased cell proliferation, survival, and metastasis. Additionally, AFP upregulates stem cell markers (CD44, CD133, EpCAM) and reprogramming proteins, supporting the expansion and maintenance of cancer stem-like cells, which are crucial for tumor initiation and progression. In addition, it inhibits apoptotic pathways (e.g., caspase-3), further enhancing tumor cell survival [146,147,148,149]. AFP also suppresses immune responses directly and indirectly via inhibition of dendritic cell maturation and function, reduction in NK and T Cell proliferation and cytotoxicity, and promotion of T reg differentiation and M2 macrophage polarization. These actions create an immunosuppressive tumor microenvironment, which allows the tumor cells to evade immune surveillance [150]. AFP shows moderate sensitivity for YSTs. Studies report that AFP is positive in about 62–89% of YST cases. AFP is highly specific for YSTs among germ cell tumors and is rarely positive in other ovarian tumors, such as clear cell carcinoma or endometrioid carcinoma. Specificity values are reported to be as high as 97.7% [151,152,153]. Glypican-3 (GPC-3) is a membrane-bound oncofetal protein that has emerged as a highly sensitive immunohistochemical marker for ovarian YST. GPC3 is positive in nearly all ovarian YSTs, with studies reporting positivity in 97–100% of cases. This makes it more sensitive than AFP, which can be negative in a subset of YSTs. GPC3 is especially valuable for distinguishing YST from clear cell carcinoma and other germ cell tumors, as it is negative in teratomas, embryonal carcinomas, and germinomas [154,155,156,157]. HNF-1β is increasingly recognized as a valuable immunohistochemical marker for YST, including ovarian YST. HNF-1β demonstrates strong nuclear staining in YST, with reported sensitivity ranging from 85% to 100% in metanalyses studies of both testicular and ovarian YSTs. HNF-1β is highly specific for YST among germ cell tumors, with specificity values between 80% and 97% [158]. It is negative in embryonal carcinoma and seminoma, and only rarely shows nuclear staining in other tumor types. While HNF-1β is robust, it is best used as part of a panel with other markers (e.g., SALL4, GPC-3, AFP) to maximize diagnostic accuracy [151]. When used together, the panel (AFP + GPC-3 + SALL4) achieves close to 100% sensitivity for YST, as nearly all YSTs will be positive for at least one marker [159,160].

Ovarian choriocarcinoma is an extremely rare and highly aggressive tumor, which can be of gestational or nongestational origin. It most commonly affects adolescents and young women, but can occur at any age. It typically appears as a solid, inhomogeneous, highly vascular adnexal mass with irregular contours and areas of hemorrhage or necrosis [161,162]. Population-based studies report a five-year relative survival rate of 89.5% for gestational choriocarcinoma, with survival exceeding 90% in localized disease and slightly lower (87.1%) in cases with distant metastasis. For nongestational choriocarcinoma (including ovarian cases), the five-year survival rate is approximately 75.5% with modern multimodal therapy (surgery and multidrug chemotherapy). Primary pulmonary choriocarcinoma and some extragonadal forms have much poorer outcomes, with five-year survival rates reported as low as below 5% [163,164,165]. β-human chorionic gonadotropin (β-hCG) is the most characteristic and specific biomarker for ovarian choriocarcinoma. Markedly elevated serum β-hCG (cut-off > 5 IU/L [166]), often in the thousands of IU/L, is a hallmark and is used for diagnosis, treatment monitoring, and recurrence detection. β-hCG is not only present in serum but also expressed in tumor tissue [98]. In case reports and small series, virtually 100% of ovarian choriocarcinoma cases show elevated β-hCG [167]. Elevated β-hCG, in combination with imaging findings (such as a highly vascular adnexal mass), strongly suggests choriocarcinoma, especially in young females or those with a relevant gynecologic history. Serial measurement of β-hCG is essential for monitoring treatment response and detecting recurrence. Declining β-hCG levels indicate effective therapy, while persistent or rising levels suggest residual or recurrent disease [168,169]. While β-hCG is highly specific for trophoblastic tumors (including choriocarcinoma), it is not exclusive to ovarian choriocarcinoma. Elevated β-hCG can also be seen in other germ cell tumors, some non-trophoblastic ovarian cancers, and in pregnancy. In a broader pediatric ovarian neoplasm cohort, β-hCG had a specificity of 76% and sensitivity of 44% for malignancy in general, but these numbers are not specific to choriocarcinoma [170,171]. Markers such as CA19-9, HE4, AFP, and LDH are sometimes measured in ovarian cancer workups, but they are not specific for choriocarcinoma and are mainly used to help differentiate between ovarian tumor subtypes [98]. A comparative analysis of biomarker performance has been listed in Table 2 below.

### 2.4. Sex Cord–Stromal Tumors

Sex cord–stromal ovarian tumors (SCSTs) are a rare, diverse group of neoplasms, accounting for about 7–8% of primary ovarian tumors. The current World Health Organization (WHO) classification divides SCSTs into three main groups: pure stromal tumors (e.g., fibroma, thecoma, sclerosing stromal tumor), pure sex cord tumors (e.g., GCT, Sertoli cell tumor), mixed SCSTs (e.g., Sertoli–Leydig cell tumor, gynandroblastoma) [176].

Ovarian fibroma is a benign, solid tumor arising purely from the stromal component of the ovary and represents about 3–4% of all ovarian neoplasms. Ovarian fibromas are mostly found in perimenopausal and postmenopausal women, though they can occasionally occur in younger patients. Ovarian fibromas are typically unilateral, firm, and well-circumscribed masses with a smooth, white, whorled-cut surface, resembling uterine fibroids. Large tumors may be associated with ascites and, rarely, Meigs syndrome [177]. Ovarian fibroma is a benign tumor, and the prognosis after treatment is excellent. The five-year survival rate for ovarian fibroma is extremely high, approaching 100% [178]. As for the time of the paper, there are no molecular or serum biomarkers uniquely distinguishing ovarian fibroma from other ovarian tumors in current clinical practice. The diagnosis of ovarian fibroma relies on histopathology and immunohistochemistry (e.g., SF-1, inhibin-α, WT1, calretinin), but these markers are not unique to fibroma and are shared with other SCSTs [179]. These four immunohistochemical markers—SF-1 (steroidogenic factor-1), inhibin-α, WT1 (Wilms tumor protein 1), and calretinin—provide critical diagnostic information for ovarian tumor classification and represent a powerful panel for distinguishing between different tumor types. SF-1 shows 100% sensitivity for ovarian SCSTs, particularly Sertoli cell tumors, while remaining negative in endometrioid carcinomas and carcinoids. Studies demonstrate that SF-1 is expressed in 100% of Sertoli cell tumors, making it superior in sensitivity to inhibin, WT1, calretinin, CD56, and CD99 for identifying steroid hormone-producing tumors [180]. Inhibin-α has become one of the most useful immunohistochemical markers for gonadal SCSTs, whether primary, recurrent, or metastatic. Inhibins are growth factors produced by ovarian follicles that regulate fertility [181]. WT1 is a tumor-suppressor gene located on chromosome 11 that plays a critical role in genitourinary system development. In ovarian pathology, WT1 serves both diagnostic and prognostic functions. It may be useful in combination with cytokreatin 7, enabling highly effective diagnosis of ovarian cancer, demonstrating its reduced expression or absence [182]. Calretinin is a 29 kDa calcium-binding protein originally discovered in neuronal tissue but also expressed in mesothelial cells and ovaries. It serves as a more sensitive but less specific marker than inhibin for SCSTs [183]. While calretinin has slightly greater sensitivity than inhibin (76% vs. 65%) for non-stromal SCSTs like GCT, it has equal specificity (92%). Calretinin is particularly useful for diagnosing SCSTs that are inhibin-negative, as it identified positivity in 32 cases that were inhibin-negative, including 2 GCT, 1 Sertoli–Leydig cell tumor, and multiple fibrous neoplasms [184]. Some research studies have explored markers related to cancer-associated fibroblasts, for example, FMO2, but these are relevant to the tumor stroma in malignant ovarian cancers, not benign fibromas [185].

GCTs account for 2–5% of ovarian malignancies. Most patients are diagnosed in their 40 s to 60 s, but cases range from adolescence to old age. The majority present at early FIGO stage I. Tumors are usually unilateral, large (mean size 10–15 cm), and may show cystic, solid, or mixed morphology. Prognosis is generally favorable, with five-year survival rates around 90–96% and mainly determined by stage at diagnosis, presence of residual tumor, and tumor size [114,186,187]. The *FOXL2 c.402C>G* (p.C134W) mutation is a hallmark of adult-type ovarian granulosa cell tumors (AGCTs). It is detected in 94–97% of adult-type GCTs, but is absent in most other ovarian and non-ovarian tumors, including juvenile GCTs and thecomas. Research reports that the *FOXL2* C134W mutation acts as a driver event in AGCT development, altering GCT and promoting tumorigenesis [188,189,190]. Mice studies revealed that introduction of the C134W mutation leads to the development of AGCTs [191]. Currently, *FOXL2* mutation analysis is a standard in AGCT diagnosis and can be performed by sequencing or in situ hybrydization. Inhibin B produced by granulosa cells is recognized as one of the most sensitive and specific serum biomarkers for ovarian GTC. It demonstrates high sensitivity (up to 93–98%) and specificity for diagnosing GCT [192,193]. Rising inhibin B levels can precede clinical or radiological evidence of recurrence by several months (2–13 months). Moreover, serum inhibin B levels correlate with tumor size and disease activity, decreasing after successful treatment and rising with progression or relapse [194]. Another important biomarker in GCT is the anti-Mullerian hormone (AMH). Meta-analyses and cohort studies show that AMH has high sensitivity (89–92%) and specificity (81–93%) for diagnosing GCT, with an area under the ROC curve of 0.92–0.93 [195,196]. AMH is produced almost exclusively by granulosa cells, making it a specific marker for GCTs, therefore, standing as a very specific marker in differentiation with other ovarian tumors. AMH performs similarly to inhibin B, and using both markers together further improves detection rates for recurrent disease as well as inhibin B; AMH levels correlate with tumor size and disease activity [197]. A unique hormonal environment created by GCTs paves the way for serum FSH as another emerging biomarker. GCTs secrete inhibin (mainly inhibin B), which acts on the pituitary to suppress FSH secretion. Preoperative serum FSH levels are significantly lower in GCT patients compared to those with other ovarian tumors. Using a cut-off of 2.0 IU/L, FSH demonstrated a sensitivity of 100% and specificity of 98% for distinguishing GCTs from other ovarian tumors, with an area under the ROC curve of 0.99. In addition, the diagnostic value of low FSH holds true regardless of whether patients are pre- or postmenopausal [198,199].

Thecomas arise from ovarian stromal theca cells and are characterized by lipid-rich, spindle-shaped neoplastic cells. They are usually unilateral and solid, with a firm, yellow-to-orange appearance on gross examination [200,201,202]. In a large population-based study, the five-year disease-specific survival rate for ovarian SCSTs (which includes thecoma and GCT) was reported as 85.7% to 89.7% [116]. Thecomas lack specific, well-established markers. Classical biomarkers such as CA-125 and HE4 panels can be used. Inhibin (especially inhibin B) is more specific for SCSTs, including GCT, but can also be elevated in thecoma, particularly in luteinized variants. However, inhibin is not routinely used for thecoma diagnosis due to its low sensitivity in this context [98].

Ovarian Sertoli–Leydig cell tumors (SLCTs) are very rare sex cord–stromal neoplasms, as they account for less than 0.5% of ovarian tumors. SLCTs typically present in young women, often with signs of androgen excess, such as hirsutism, deepening of the voice, or amenorrhea. The five-year survival rate is 92–100% for patients with disease confined to the ovary or pelvis. The five-year survival rate drops to 33–80% for patients with extra-ovarian spread. In a large SEER database study, the overall 5-year cancer-specific survival for malignant SLCTs was 76.2% [141,203,204]. No single immunohistochemical marker is absolutely specific for SLCTs alone, though *DICER1* mutation is the most specific molecular marker for moderately and poorly differentiated ovarian Sertoli–Leydig cell tumors. Studies report that 88–100% of these tumors harbor either somatic or germline *DICER1* mutations, especially in the RNase IIIb domain [205,206]. On the other hand, well-differentiated SLCTs rarely have *DICER1* mutations [207]. Most SLCTs also stain positively for inhibin A, which is particularly useful in differentiating SLCTs from other ovarian tumors with similar morphology, such as small cell carcinoma or Sertoliform endometrioid carcinoma. However, it cannot distinguish between different types of SCSTs [208,209]. SF1 also shows strong nuclear staining in nearly all SLCTs, though as well as inhibin A, it is characteristic of sex cord–stromal origin, not the SCLT itself. Studies report that SLCTs, along with other SCSTs, show strong and diffuse nuclear staining for SF1 in over 75% of cases, regardless of differentiation grade [210,211]. Table 3 presents an analysis of biomarker performance across major SCST subtypes. Table 4 presents a summary of key findings on some presented biomarkers.

## 3. Novel Potential Biomarkers

### 3.1. MiRNAs in Ovarian Cancer

MiRNAs are currently considered to be among the most promising biomarkers in the context of ovarian cancer diagnosis and treatment. The diversity of miRNA expression profiles between different histological types of cancer allows them to be used both in the detection and precise subclassification of ovarian tumors. For example, the miR-200 family (miR-200a, miR-200b, miR-200c, miR-429) is characterized by significantly increased expression in serous ovarian cancers, while miR-192 and miR-215 are specific to mucinous cancers. Such characteristic miRNA expression patterns not only allow for the differentiation between benign and malignant lesions, but also for the identification of specific histological subtypes, which plays a key role in the selection of a personalized therapeutic strategy [217,218,219]. The expression of selected miRNAs, such as further let-7g type and the aforementioned miR-200c, shows a significant correlation with the sensitivity of cancer cells to chemotherapy, especially to treatment based on platinum derivatives, such as cisplatin. Decreased levels of let-7g are associated with the development of resistance to chemotherapy and poor clinical prognosis, indicating the usefulness of miRNA both as a diagnostic biomarker and as a tool for monitoring the effectiveness of anticancer therapy [218,219,220]. The miR-200 family is a critical molecular regulator in the pathogenesis of ovarian cancer, functioning as an epithelial-to-mesenchymal transition (EMT) suppressor. This action is achieved through the post-transcriptional silencing of the transcription factors *ZEB1* and *ZEB2*, which are key EMT inducers. The direct suppression of *ZEB1/ZEB2* expression by miR-200 results in the maintenance of high levels of the adhesion molecule E-cadherin and the stabilization of an epithelial phenotype in cancer cells. A decrease in miR-200 expression is correlated with the induction of EMT, leading to an increase in invasiveness and metastatic potential in ovarian cancer cells. Conversely, maintaining high levels of miR-200 restricts cell dispersion and metastasis, suggesting its value as a prognostic and diagnostic biomarker and implying its potential for use in targeted therapy. It should be emphasized that the role of miR-200 is context-dependent and integrated within a regulatory network, also influencing the chemosensitivity of tumor cells and other key hallmarks of malignancy [221,222]. Exosomal miRNAs are promising non-invasive biomarkers. They can be used to monitor treatment response and assess the risk of cancer recurrence. The profiling of miRNAs isolated from the serum or plasma of patients with ovarian tumors shows high diagnostic sensitivity and specificity, enabling effective detection of ovarian cancer, even in its early stages. This is of particular clinical importance given the often-asymptomatic course of the disease at the initial phases of carcinogenesis, which makes early diagnosis using conventional methods difficult [223,224,225].

### 3.2. Lipid Metabolism Alterations, Metabolomic and Proteomic Biomarkers

Modifications in the lipid profile are an important element of the metabolic reprogramming of cancer cells in ovarian cancer, contributing to the intensification of their growth, invasive and metastatic abilities, and the development of resistance to treatment. Cancer cells exhibit an enhanced pathway of biosynthesis and accumulation of free fatty acids, triglycerides, phospholipids and cholesterol, which promotes increased proliferation and migration of tumor cells. Elevated levels of unsaturated fatty acids and increased activity of desaturating enzymes, such as SCD1 and FADS2, support the maintenance of the tumor stem cell phenotype and the development of resistance to chemotherapy [226].

The lipid profile of ovarian cancer patients is characterized by a global decrease in the concentrations of most serum lipid classes, with a simultaneous increase in the levels of selected ceramides and triglycerides. These changes correlate with disease progression and may exceed the classic CA-125 marker in terms of prognostic value. Lipid metabolism disorders are already observed in the early stages of cancer, and their intensity increases with its progression, regardless of the histopathological type [227].

Furthermore, the reprogramming of lipid metabolism affects the tumor microenvironment, promoting the development of immunosuppression and enabling the tumor to evade the host’s immune response. Consequently, the deregulation of lipid pathways is not only a characteristic feature of ovarian cancer biology, but also a promising diagnostic, prognostic, and therapeutic biomarker in this disease entity [228].

Metabolomic clinical research studies have revealed significant abnormalities in lipid profiles in patients with ovarian cancer, with a particularly significant reduction in lysophosphatidylcholine (LPC) and other phospholipid concentrations in plasma compared to healthy individuals. These changes are most pronounced in cases of HGSOC. At the same time, an increase in the concentrations of selected lipids and metabolites, including hydroxybutyrate derivatives, is observed in tumor tissues, which leads to another group of biomarkers—metabolites, especially lipids, the presence of which correlates with advanced disease stage and poor prognosis. The accumulation of these compounds is associated with increased migration and invasiveness of cancer cells, as well as a decrease in the expression of enzymes involved in hydroxybutyrate metabolism, which may be a potential prognostic indicator [229,230,231]. Proteomics, as a field dealing with the comprehensive analysis of the protein profile of cells and body fluids, enables the identification of specific protein panels characteristic of different types of ovarian tumors. In the case of serous ovarian cancers, significant differences in the expression of over 200 proteins have been demonstrated in comparison with benign and borderline malignant tumors. Among the proteins with increased expression in malignant tumors are several tissue-based markers associated with disease development and progression. Argininosuccinate synthetase 1, involved in the urea cycle and arginine biosynthesis, is overexpressed in HGSCA. Its upregulation facilitates tumor growth and metastasis and may support the increased metabolic demands of rapidly proliferating cancer cells. PPA1 (inorganic pyrophosphatase) is also upregulated, and its overexpression is linked to poor survival and chemoresistance. BCAT1 (branched-chain amino acid transaminase) helps differentiate SCA from benign tumors. It also promotes proliferation and invasion in epithelial ovarian cancer and correlates with poor clinical outcomes. MCM4 (minichromosome maintenance complex component 4) similarly assists in distinguishing SCA from benign lesions. Its expression is closely tied to cancer stages and shows high predictive value in ovarian cancer diagnostics. In contrast, several proteins show decreased levels. MUC5B (mucin-5B), a glycoprotein variably expressed in gastric and breast cancers, is reduced in higher-grade ovarian tumors. SLC4A1 (solute carrier family 4 member 1) is also downregulated and is associated with poor progression-free survival. Tenasin-XB (TNXB), another downregulated marker, is linked to disease progression in SCA. In addition, certain proteins, such as *WFDC2* (also known as HE4) (WAP four-disulfide core domain protein 2/Human Epididymis Protein 4—used in triaging or early diagnosis of ovarian cancer) or KRT19 (keratin 19, a protein that is part of the intermediate filament system and helps maintain the structural integrity of epithelial cells) are highly sensitive and specific in differentiating malignant from benign lesions, and their associations can be successfully used in predictive models to support the diagnostic process [232,233,234,235]. An integrated analytical approach combining metabolomic and proteomic analyses offers the possibility of not only earlier detection of ovarian cancer, but also of identifying biomarkers predictive of response to treatment, including markers associated with resistance to chemotherapy. Specific protein and metabolic profiles may indicate an increased risk of disease recurrence or progression, which provides the basis for a more precise, personalized selection of therapeutic strategies.

In turn, proteogenomics—integrating information from proteomic and genomic analyses—enables the identification of new molecular targets for therapy and more effective stratification of patients, for example, in the context of eligibility for treatment with PARP inhibitors or therapies targeting signaling pathways associated with mechanisms of resistance to chemotherapy [233,235,236,237,238].

### 3.3. ctDNA

ctDNA is becoming increasingly important as a biomarker in the diagnosis, monitoring and personalization of ovarian cancer treatment. The detection of ctDNA after surgery is a strong indicator of decreased recurrence-free survival [239]. ctDNA analysis enables non-invasive real-time tracking of genetic changes, allowing for the assessment of treatment efficacy and early detection of therapeutic resistance, especially in the context of advanced and recurrent forms of ovarian cancer. The high degree of concordance between mutations detected in ctDNA and tumor tissue, particularly in cases of HGSOC, confirms the clinical usefulness of this tool, creating a new approach in the field of biomarkers. In HGSOC, the detection of mutations in the *TP53*, *BRCA1/2* and *TP53BP1* genes in ctDNA allows the prediction of response to treatment with PARP inhibitors and platinum-based chemotherapy. In addition, analysis of ctDNA levels allows monitoring of treatment efficacy and risk of recurrence, with changes in ctDNA concentration often preceding traditional markers such as CA-125 in identifying disease progression. The decrease in ctDNA concentration observed after treatment correlates with a favorable prognosis, while its persistence or increase may indicate the presence of residual disease or the development of resistance to treatment. However, it is worth noting that the diagnostic effectiveness of ctDNA as a biomarker is the highest in HGSOC, while in other histological types, such as CCOC or MOC, the sensitivity of detection and the concordance of mutations with tumor tissue are significantly lower, which limits the use of this method in these patient groups. The use of advanced next-generation sequencing (NGS) technologies enables the identification of a wide spectrum of genetic and epigenetic changes in ctDNA, which opens up new perspectives for the precise stratification of patients and the selection of targeted therapies for the treatment of ovarian cancer. [240,241,242,243,244,245,246].

### 3.4. Transcriptomic Biomarkers

Gene expression analysis and transcriptomic studies are important tools in identifying biomarkers for ovarian cancer, contributing to the precise differentiation of histopathological types, the prediction of clinical disease progression, and the development of personalized treatment strategies. Highly sensitive gene panels include components of the Notch and Wnt signaling pathways. The Notch pathway, which regulates normal embryonic development and tissue homeostasis and is involved in carcinogenesis, encompasses receptors NOTCH1–NOTCH4, the ligands DLL1 and JAG2, and HES1, a marker that can indicate cancer progression. The Wnt pathway includes CTNNB1, the gene-encoding β-catenin, which regulates cell growth, renewal, and differentiation. All of these are proteins that regulate stemness, differentiation, and proliferation in fallopian tube cells. They also serve as biomarkers of HGSOC. Gene panels incorporating these markers show high diagnostic efficacy, reaching up to 100% sensitivity and specificity in some models in distinguishing HGSOC from benign lesions. The expression of genes belonging to the above pathways is significantly reduced in cases of malignant tumors, which shows a significant correlation with the clinical stage of the disease and the concentration of markers such as CA-125. Furthermore, other studies suggest that increased expression of genes such as *KRAS*, *c-FOS*, *PUMA* and *EGFR* may be characteristic of ovarian cancer and associated with a poor prognosis [247,248,249].

Transcriptomics enable comprehensive identification of a wide range of coding genes and long non-coding RNAs (lncRNAs) whose expression shows significant correlations with clinical parameters such as patient age at diagnosis, lymphatic invasion, residual tumor mass and resistance to chemotherapy. Selected lncRNA and mRNA transcripts can serve as molecular prognostic indicators and form the basis for constructing predictive models that enable the assessment of disease recurrence risk and the prediction of overall survival in ovarian cancer patients [250]. Differentiation in the expression of selected genes, such as *ABCG2*, *DOCK4*, *DUSP1/4/5*, *GADD45B*, *HELQ*, *HOXA9*, *KLF4*, and *NR4A1*, enables the identification of patients who are sensitive or resistant to platinum-based chemotherapy. The obtained transcriptomic data provide a potential basis for the development of individualized treatment strategies and can serve as a tool for the early detection of drug resistance mechanisms, which is important for improving the effectiveness of cancer therapy [251]. Table 5 presents a summary of findings on some presented novel biomarkers.

## 4. Clinical Applications of Biomarkers

The CA-125 and HE4 tumor markers are widely used for diagnosis, treatment monitoring, and recurrence risk assessment. Their concentrations provide important prognostic and therapeutic information, enabling the effectiveness of the treatment used to be assessed, although they do not constitute a clear basis for choosing a specific treatment regimen. The use of the ROMA algorithm, which integrates CA-125 and HE4 levels, significantly increases the accuracy of diagnosis and allows for more effective risk stratification, especially in postmenopausal patients, which may influence clinical decisions regarding the extent of surgical intervention and the intensity of oncological treatment [37,98,252]. AFP, β-hCG and LDH markers are important for diagnosis and treatment monitoring of OGCTs, but their role in treatment individualization is limited to response and prognosis assessment. [98,253]. MiRNA and ctDNA profiling enables the detection of chemotherapy resistance, monitoring of minimal residual disease, and dynamic adjustment of therapy to molecular changes in the tumor. Proteomic and metabolomic analysis allows for the identification of cancer subtypes and the selection of targeted therapies, while gene expression and transcriptomics can indicate the presence of mutations that are predisposed to the effectiveness of specific drugs, e.g., PARP inhibitors [37,98,254,255]. In clinical practice, immunoenzymatic assays (ELISAs) are most commonly used to determine classic tumor markers such as CA-125, HE4, AFP, β-hCG and LDH. These methods are widely available, quick to perform and relatively inexpensive, which allows for their routine use in diagnosis and treatment monitoring. Diagnostic accuracy can be further improved by using multiparametric algorithms, such as ROMA or RMI, which integrate biomarker results with clinical data, such as the patient’s menopausal status or ultrasound image, improving the sensitivity and specificity of the diagnosis [36,37,256]. In the case of modern biomarkers, such as miRNA, ctDNA, DNA methylation or gene expression, advanced molecular techniques are used: PCR, qPCR, NGS, microarrays and mass spectrometry (in proteomics and metabolomics) [224,254,257,258]. CtDNA methylation analysis is most often based on methylation-specific PCR, while proteomic studies use methods such as iTRAQ, TMT or MS [254,257]. Biosensors and artificial intelligence-based platforms are also becoming increasingly important, enabling rapid, automated and sensitive detection of biomarkers in body fluids [259,260]. The availability of biomarker tests covering classic markers (CA-125, HE4, AFP, β-hCG, LDH) and complex diagnostic algorithms (such as the ROMA score), as well as modern molecular tools (miRNA profiling, metabolite analysis, proteomics, ctDNA, gene expression and transcriptomics) varies greatly depending on the type of biomarker, geographical region and level of healthcare system development. The most commonly used markers in routine clinical practice remain CA-125 and HE4, which are widely available in Europe, North America and Asia. Their diagnostic effectiveness in relation to ovarian cancer has been confirmed by numerous clinical studies and meta-analyses. However, it is worth noting that CA-125 levels have limited sensitivity in detecting early stages of the disease and may also be elevated in non-cancerous conditions such as menstruation or endometriosis. HE4, on the other hand, has higher specificity and does not increase in the case of endometriosis, making it a valuable addition to the diagnostic panel. Diagnostic accuracy can be further improved by using the ROMA algorithm, which integrates CA-125 and HE4 levels while taking into account the patient’s menopausal status. This algorithm is particularly accurate in differentiating between benign and malignant lesions in postmenopausal women, although its availability depends on the implementation of appropriate laboratory analyses [36,96,256,261,262]. AFP, β-hCG and LDH markers are routinely used in the diagnosis of OGCTs and liver cancer, and are globally available in both developed and developing countries. They are widely used in clinical practice, although their sensitivity and specificity are limited, especially in the early stages of the disease [98]. Modern molecular tests, such as miRNA, ctDNA, metabolite analysis, proteomics, gene expression and transcriptomics, are currently mainly available in scientific research or in selected reference centers, mainly in areas with a high level of innovation, such as the USA, Japan and Western Europe. These tests show potential for increasing the sensitivity and specificity of diagnostics, as well as for personalizing treatment, but their routine use is limited by high costs, lack of standardization and the need for further clinical validation [98,256,261]. Similarly, tests based on ctDNA or advanced proteomics and metabolomics, such as LPC or glycoprotein determination, are promising, but their availability is limited to scientific research and selected laboratories [96,256].

## 5. Challenges, Limitations, and Future Perspectives

Despite significant advancements in the identification of ovarian cancer biomarkers, their implementation and widespread use in clinical practice face several obstacles. Among the various challenges, one of the most critical limitations is the biological heterogeneity of ovarian tumors. This heterogeneity, stemming from epigenetic variations, environmental influences, and genetic mutations, affects not only tumor progression but also treatment efficacy, disease recurrence, and the accuracy of diagnostic processes in determining the specific tumor type [263,264]. Current methods for assessing heterogeneity primarily rely on tumor tissue biopsies and sequencing to obtain a clonal expansion score. However, this approach is highly complex, labor-intensive, and difficult to incorporate into routine clinical practice. Emerging evidence suggests that sequencing ctDNA represents a promising, minimally invasive alternative for evaluating tumor heterogeneity. Nevertheless, ctDNA analysis may not fully and accurately reflect the genomic landscape of ovarian tumors [265]. Another major limitation lies in the low diagnostic performance of single biomarkers. The concentrations of these biomarkers are influenced by a range of factors, including menopausal status and comorbidities such as endometriosis. For instance, CA-125 and HE4 levels vary significantly among patients due to the influence of such confounding variables [98,266,267]. Studies have demonstrated that combinations of multiple biomarkers yield markedly higher specificity and sensitivity compared to individual markers. Mao et al. reported encouraging results in this clinical research regard; however, the authors emphasized the need for further investigations, as their study was limited by a relatively small sample size and low demographic diversity among participants [268]. Genomic and proteomic profiling-based tests offer promising avenues for identifying potential ovarian cancer biomarkers, but they are not without limitations. Findings by Arad et al. revealed discrepancies between protein expression and RNA levels, underscoring the challenges of accurately predicting protein expression solely based on RNA data. The authors also highlighted substantial gaps in proteomic research, particularly the lack of sufficient data on patient prognosis and treatment response [269,270]. Furthermore, the implementation of such advanced diagnostic techniques is associated with high costs, which may hinder their adoption, especially in healthcare systems with limited resources or varying reimbursement policies across different countries [271]. Another critical challenge is low public awareness regarding ovarian cancer symptoms and the importance of regular screening. Hong et al., in their metanalyses, emphasized the vital role of public health campaigns in disseminating knowledge about both the disease itself and the available diagnostic methods [272].

Despite persistent issues such as disease recurrence and treatment resistance among patients with advanced-stage ovarian cancer, the outlook for future detection and treatment strategies remains promising. A multimarker approach, combined with integrated data from proteomics, genomics, and transcriptomics—collectively referred to as a multi-omics strategy—offers a more comprehensive understanding of the disease. The incorporation of artificial intelligence (AI), machine learning, and advancements in nanomedicine is expected to pave the way for more precise diagnostic tools, advanced imaging techniques, and highly personalized treatment modalities [264,273]. AI can be employed to analyze vast biomarker datasets, predict disease progression, and evaluate the potential success of various therapeutic strategies, directly contributing to the development of precision medicine [259]. However, the integration of AI into oncology requires the establishment of robust systems capable of securely storing, analyzing, and protecting sensitive patient data. Ensuring such infrastructure, while also providing equitable access to these novel technologies, remains a significant challenge due to economic disparities worldwide [274]. Advancements in nanotechnology and biosensor development are also driving progress in diagnostic capabilities. Modern biosensors based on two-dimensional MXene nanomaterials exhibit exceptional sensitivity, enabling the detection of extremely low concentrations of biomarkers. This breakthrough indicates the potential of these devices to revolutionize ovarian cancer diagnostics [275].

In summary, the most significant barriers to effective ovarian cancer management and optimal treatment selection include tumor biological heterogeneity, limited performance of current screening methods, challenges with standardizing and validating detection techniques, and a pronounced shortage of large, multicenter studies involving diverse demographic populations. Thorough reviews of such studies could serve as a foundation for the development of unified diagnostic and therapeutic protocols. To achieve these goals, there is a critical need for greater data collection, comprehensive analyses, and systematic training of healthcare professionals to enhance the accuracy and efficacy of diagnostic and treatment tools. An interdisciplinary approach is essential for advancing precision medicine and improving patient outcomes. The rapid progress in research over recent and forthcoming years offers a strong opportunity to overcome current limitations in ovarian cancer diagnosis and therapy. Nonetheless, there remains an urgent need for global standardization of ovarian cancer biomarkers, and we strongly advocate for intensified international collaboration to address this pressing challenge.

Take-home messages:-CA-125 and HE4 remain the most commonly used biomarkers in clinical practice due to their wide availability. They provide essential diagnostic, prognostic, and monitoring information, particularly when combined within the ROMA algorithm.-Modern molecular biomarkers—including miRNAs, circulating tumor DNA (ctDNA), proteomics, metabolomics, and gene expression profiles—hold significant potential for the development of personalized treatment strategies.-However, their current use remains largely confined to research settings and specialized centers. The implementation of novel biomarkers into routine clinical practice is limited by high costs, lack of standardization, and restricted accessibility, underscoring the need for further validation studies prior to their widespread global adoption.

## Figures and Tables

**Figure 1 ijms-26-11702-f001:**
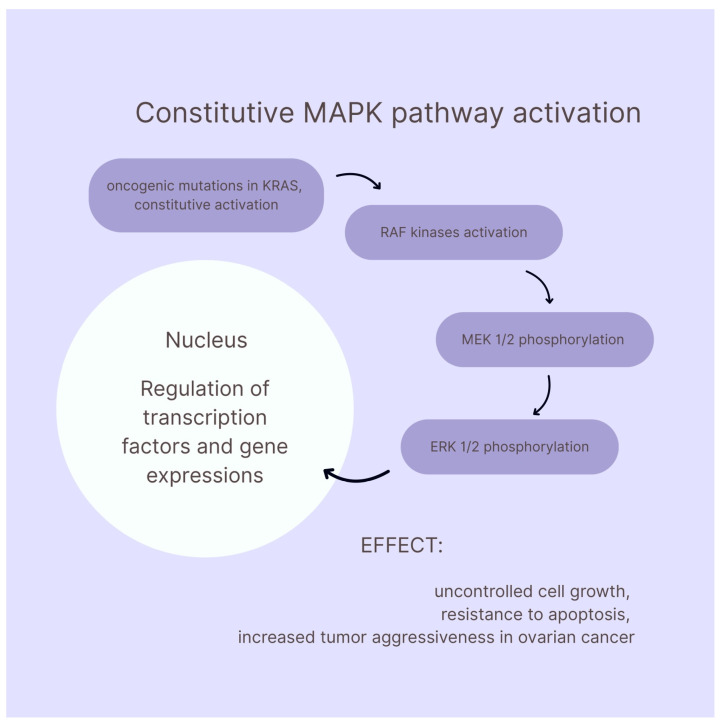
Constitutive MAPK pathway activation.

**Table 1 ijms-26-11702-t001:** Summary of biomarker classes performance for particular subtypes of epithelial ovarian cancer.

Subtype	Serum Markers	IHC Panels	Genetic/Transcriptomic	Circulating Biomarkers	Proteomic/Metabolomic	References
HGSOC	Moderate (improved in panels)	High (WT1, p53, multimarker)	High (*TP53*, *BRCA1/2*, miRNAs)	High (ctDNA, miRNAs)	High (panels, metabolomics)	[95,96,97,98,101,103,104]
CCOC	Low	High (Napsin A, HNF-1β)	Moderate (*ARID1A*, methylation)	Emerging	Emerging	[98,106]
MOC	Low	Moderate (CK7, CK20)	Moderate (*KRAS*)	Emerging	Emerging	[96,98]
EOC	Moderate (best in panels)	High (multimarker)	High (methylation, gene panels)	Moderate	High (panels)	[96,98,106]
LGSC	Low	Moderate (WT1, ER, PR)	Moderate (*KRAS*, *BRAF*)	Emerging	Emerging	[96,99,106]

**Table 2 ijms-26-11702-t002:** Summary of optimal biomarkers for each OGCT subtype.

Subtype	Best Serum Biomarker(s)	Key IHC Marker(s)	Molecular/Genetic	Circulating miRNA	References
Dysgerminoma	LDH	SALL4, OCT4, PLAP	KIT (subset)	miR-371a-3p	[123,124,172,173,174]
Yolk Sac Tumor	AFP	SALL4, GPC3, HNF-1β	Isochromosome 12p	miR-371a-3p	[123,124,151,172,173,174]
Choriocarcinoma	β-hCG	SALL4	-	miR-371a-3p	[173,175]
Teratoma	None	SALL4 (immature only)	-	Not expressed	[123,124,172,175]

**Table 3 ijms-26-11702-t003:** Comparative biomarker performance across major SCST subtypes.

Subtype	Key Serum Markers	Key IHC Markers	Key Genetic Alterations	Diagnostic/Prognostic Notes	References
Adult GCT	Inhibin B, AMH	SF-1, inhibin-α, calretinin, FOXL2	*FOXL2 C134W*, *TERT* promoter	*FOXL2* mutation is diagnostic; inhibin B/AMH best for monitoring; *TERT* mutation = worse prognosis	[180,193,212,213,214,215]
Juvenile GCT	Inhibin B, AMH	SF-1, inhibin-α, calretinin	*DICER1* (some cases)	*DICER1* mutations in subset; similar IHC to AGCT	[206,207,214]
SLCT	Inhibin B	SF-1, inhibin-α, calretinin, FOXL2	*DICER1* (most), *FOXL2* (rare), *TERT* (rare)	*DICER1* mutations in most; IHC panel highly sensitive	[180,206,207,211,212,216]
Thecoma	Inhibin B (variable)	SF-1, inhibin-α, calretinin	Non-specific	IHC confirms diagnosis; serum markers less reliable	[214]

**Table 4 ijms-26-11702-t004:** Summary of key findings on some presented biomarkers.

Biomarker	Tumor Type	Implication	Sensitivity/Specificity	Clinical Significance	References
CA-125	HGSOC, EOC	For screening and disease monitoring	63–83%/71–83%	Well-established, used alone shows limited value, enhanced in combination with HE4	[36,37,49,98]
HE4	HGSOC, EOC	Most effective for differentiating between benign and malignant masses	64–75%	A superior tool compared with CA-125, though the combination of both and Smac protein yields optimal sensitivity, but low specificity	[36,37,49,50,98]
WT1, PAX8	HGSOC	For screening	>90%/low specificity	Beneficial in the process of diagnostic differentiation	[44]
IMP3, Napsin A, HNF-1β	CCOC	Compromised as a panel	>80%/>80%	Beneficial in the process of diagnostic differentiation	[57]
IFITM1	CCOC	For disease monitoring/prognostic	-	It is one of the proteins that is correlated with recurrence-free survival	[63]
CK7+, SATB2-	MOC	Role in distinguishing primary ovarian mucinous tumors from colorectal and appendiceal metastases	78%/99%	The most critical panel in MOC that outperforms traditional panels	[70,71,72]
SALL4	GCT	For disease detection and diagnostic differentiation	73%/high specificity	It is consistently expressed in germ cell tumors, only rarely observed in non-germ cell tumors, and absent in mature teratomas	[119,124]
OCT4	Most dysgerminomas	For disease detection and diagnostic differentiation	-	OCT4 is highly specific, because all or nearly all dysgerminomas are OCT4-positive	[133,134,135]
AFP	YST	For disease detection and diagnostic differentiation.	Moderate/97.7%	All ovarian YSTs show markedly elevated serum AFP levels, while they are rarely positive in other ovarian tumors	[144,145,152,153]
GPC3	YST	For disease detection and diagnostic differentiation.	-	especially valuable for distinguishing YST from clear cell carcinoma and other germ cell tumors, as it is negative in teratomas, embryonal carcinomas, and germinomas	[154,155,156,157]
β-hCG	choriocarcinoma	For diagnosis, treatment monitoring, and recurrence detection	-	Elevated β-hCG, in combination with imaging findings, strongly suggests choriocarcinoma	[167,168,169]
Inhibin B	GCT, thecoma	For diagnosis and disease monitoring	89–98%/81–93%	Beneficial in the process of diagnostic differentiation, both Inhibin B and AMH together further improve detection rates for recurrent disease	[192,193,195,196]

**Table 5 ijms-26-11702-t005:** Summary of key findings on some presented novel biomarkers.

Marker Category	Specific Marker/Feature	Biological Role/Correlation	Clinical Application	References
miRNAs	miR-200 family (miR-200a, miR-200b, miR-200c, miR-429)	Significantly increased expression in serous ovarian cancers. Acts as an EMT suppressor by silencing *ZEB1*/*ZEB2*. High levels restrict metastasis.	Histological subclassification (serous subtype). Prognostic biomarker. Potential therapeutic target.	[218,219,220,221,222,223,224,225]
	miR-192, miR-215	Specific to mucinous cancers.	Histological subclassification (mucinous subtype).	
	let-7g, miR-200c	Decreased levels of let-7g correlate with chemotherapy resistance (platinum derivatives) and poor prognosis.	Monitoring treatment effectiveness. Prognostic biomarker.	
	Exosomal miRNAs	High diagnostic sensitivity and specificity in serum/plasma.	Non-invasive diagnosis (even in early stages). Monitoring treatment response and recurrence risk.	
ctDNA	ctDNA level/mutations (*TP53*, *BRCA1/2*, *TP53BP1*)	High concordance with tumor tissue (especially in HGSOCs). Decrease after treatment correlates with favorable prognosis. Mutations predict response to PARP inhibitors and platinum-based chemotherapy.	Monitoring treatment efficacy (changes often precede CA-125). Predicting therapeutic response. Early detection of resistance.	[240,241,242,243,244,245,246]
	Diagnostic effectiveness	Highest in HGSOCs. Lower sensitivity in CCOC and MOC.	Stratification of patients. Methodological limitations in non-HGSOC types.	
Transcriptomic Markers	Gene panels (Notch, Wnt pathways)	Expression is significantly reduced in malignant tumors (high diagnostic efficacy, up to 100% sensitivity/specificity for HGSOCs vs. benign).	Precise differentiation of HGSOCs from benign lesions. Correlation with clinical stage (CA-125).	[247,248,249,250,251]
	Increased expression genes	*KRAS*, *c-FOS*, *PUMA*, *EGFR* associated with poor prognosis.	Prognostic indicator.	
	Chemoresistance markers	Differential expression of *ABCG2*, *DOCK4*, *DUSP1/4/5*, *HELQ*, *HOXA9*, *KLF4*, etc.	Identifying sensitivity/resistance to platinum-based chemotherapy. Basis for personalized treatment.	
Lipid Metabolism Alterations	Enzymes (SCD1, FADS2)	Increased activity supports the tumor stem cell phenotype and chemoresistance.	Potential therapeutic targets.	[226,227,228]
	Serum/Plasma Lipid Profile	Global decrease in most lipid classes, concurrent increase in selected ceramides and triglycerides. Significant reduction in LPC (especially HGSOC).	Correlates with disease progression. Prognostic value may exceed CA-125. Early-stage detection.	
	Tissue Metabolites	Increased hydroxybutyrate derivatives in tumor tissue correlate with advanced stage and poor prognosis.	Prognostic indicator.	
Proteomic Markers	*WFDC2* (HE4), *KRT19*, *RBFOX3*	Highly sensitive and specific for differentiating malignant from benign lesions.	Predictive models to support the diagnostic process.	[229,230,231,232,233,234,235,236,237,238]
	Proteins with increased/decreased expression (in serous cancers)	Increased: Argininosuccinate synthetase 1, PPA1, BCAT1, MCM4. Decreased: MUC5B, SLC4A1, TNXB.	Identifying specific protein panels for tumor types.	
Integrated Markers	Proteogenomics (Proteomic + Genomic)	Integrates information for targeted therapy.	Identification of new molecular targets. Effective patient stratification (e.g., eligibility for PARP inhibitors).	

## Data Availability

No new data were created or analyzed in this study. Data sharing is not applicable to this article.

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
