# Peer review of "Ovarian Tumor Biomarkers: Correlation Between Tumor Type and Marker Expression, and Their Role in Guiding Therapeutic Strategies"

_ijms, 2025, doi:10.3390/ijms262311702_

Round 1
Reviewer 1 Report
Comments and Suggestions for Authors
Journal: IJMS (ISSN 1422-0067)
Manuscript ID: ijms-3914421
Type: Review
Title: Ovarian Tumor Biomarkers: Correlation Between Tumor Type and Marker Expression, and Their Role in Guiding Therapeutic Strategies
The manuscript provides a thorough and well-organized review of ovarian tumor biomarkers, effectively covering both classical and emerging markers, with a focus on their diagnostic, prognostic, and predictive value. The authors have done an excellent job of compiling a diverse body of literature and presenting complex clinical and molecular data in a clear and understandable manner. The inclusion of histological subtypes and associated biomarkers provides significant value to clinicians and researchers alike. The manuscript also successfully integrates traditional biomarker knowledge with modern approaches such as microRNAs, ctDNA, metabolomics, and proteomics, while emphasizing future directions and clinical applications.
Minor revision needed:
1: Introduction is descriptive, resembling a textbook, and would benefit from a more focused problem statement.
2: Only 20–25% of patients receive an early diagnosis, but there is no comparison with a global dataset. It is recommended to include data from WHO or GLOBOCAN for more comprehensive insights.
3: Line 248–253: SALL4 is identified as a biomarker for immature teratomas; however, its function in maintaining stemness and pluripotency is unclear. It is suggested to add a mechanism explaining how SALL4 influences the undifferentiated phenotype.
4: Line 300–305: AFP is discussed as a biomarker for immature teratomas (YSTs), yet its mechanism in cell proliferation, immune suppression, and pluripotency maintenance remains unexplained.
5: Line 195–200: KRAS and HER2 mutations in malignant ovarian cancer (MOC) are noted, yet there is a lack of mechanistic explanation regarding how KRAS activation stimulates MAPK pathway signaling. A pathway diagram is requested if available.
Author Response
Comment 1: Introduction is descriptive, resembling a textbook, and would benefit from a more focused problem statement.
Response 1: Thank you for this valuable suggestion. The Introduction section has been revised and rewritten to provide a more focused problem statement and to better highlight the research objectives
Comment 2: Only 20–25% of patients receive an early diagnosis, but there is no comparison with a global dataset. It is recommended to include data from WHO or GLOBOCAN for more comprehensive insights.
Response 2: Thank you for this valuable comment. The sentence has been revised accordingly, and global
epidemiological data from GLOBOCAN have been incorporated to provide a broader and more
comprehensive context for comparison.
Comment 3: Line 248–253: SALL4 is identified as a biomarker for immature teratomas; however, its function in maintaining stemness and pluripotency is unclear. It is suggested to add a mechanism explaining how SALL4 influences the undifferentiated phenotype.
Response 3: We wholeheartedly support the need for explanation of mechanisms standing behind SALL4. We enriched the text with proper explanation, and we hope that the updated version enhances the
understanding of why SALL4 is crucial for maintaining stemness and pluripotency, therefore
producing a more relevant context for further discussion.
Comment 4: Line 300–305: AFP is discussed as a biomarker for immature teratomas (YSTs), yet its mechanism in cell proliferation, immune suppression, and pluripotency maintenance remains unexplained.
Response 4: Thank you for indicating lack of the explanation, it undoubtedly hinders the reader from
understanding the fundamental role of AFP in prognosis of YST. We developed the above section,
with a view to making the role of AFP in YSTs more transparent and understandable.
Comment 5: Line 195–200: KRAS and HER2 mutations in malignant ovarian cancer (MOC) are noted, yet there is a lack of mechanistic explanation regarding how KRAS activation stimulates MAPK pathway signaling. A pathway diagram is requested if available.
Response 5: The mechanism has been described, with an appropriate pathway diagram added to provide more clear visualization.
Reviewer 2 Report
Comments and Suggestions for Authors
This manuscript provides a comprehensive and up-to-date review of biomarkers in ovarian cancer, with an emphasis on emerging diagnostic and predictive tools such as microRNAs, ctDNA, metabolomics, proteomics, and transcriptomics. It effectively summarizes the current clinical applications, technological advances, and future challenges in the field. The text is well-structured, following a logical flow from established biomarkers to novel ones, clinical applications, and future perspectives.
The scientific value of this work is considerable, as it provides clinicians and researchers with a synthesized overview of the field and identifies areas requiring further research, especially regarding standardization, sensitivity, and accessibility of novel biomarkers.
However, several areas need to be strengthened, particularly in terms of critical analysis, language clarity, and integration of mechanistic insights.
Major Concerns
1. The review is informative but remains largely descriptive, focusing on listing biomarkers and their clinical implications without sufficient critical comparison of their relative performance and readiness for clinical use. Table 1 provides useful information on classical biomarkers but does not cover emerging biomarker classes (e.g., microRNAs, ctDNA, proteomics, metabolomics) or contrast their strengths, weaknesses, and validation status. Consider expanding the existing table or adding a comparative summary that includes sensitivity, specificity, clinical validation stage, advantages, and limitations of each biomarker class to enhance the analytical depth of the review.
2. The manuscript summarizes biomarker data effectively but provides little mechanistic explanation linking these biomarkers to ovarian cancer pathophysiology. For example, the section on microRNAs could explain how the miR-200 family affects EMT pathways or how lipid profile changes relate to tumor metabolism. My suggestion is to briefly integrate molecular pathways or mechanistic models that explain why these biomarkers change in ovarian cancer. Integrating even brief mechanistic insights would elevate the review from a descriptive summary to a more analytical and impactful contribution.
3. The text acknowledges the heterogeneity of ovarian tumors but does not systematically discuss how different biomarker classes perform in specific histological subtypes (e.g., HGSOC vs. CCOC vs. MOC). Expand on subtype-specific biomarker utility — this is particularly important for clinical translation.
4. Many statements rely on clusters of references, but the source quality, study type, and strength of evidence are not clear. Where possible, specify whether cited studies are meta-analyses, clinical trials, or basic research to strengthen the manuscript’s scientific rigor.
Minor Concerns
1. Several sentences are too long or repetitive, which can reduce readability.
Example: “MicroRNAs present in exosomes are promising, non-invasive biomarkers that can be used both to monitor response to treatment and to assess the risk of cancer recurrence.” → Could be shortened for clarity. It is better to simplify and split long sentences.
2. Terms like “ctDNA,” “HGSOC,” “MOC,” and “CCOC” are used without a consistent prior definition. Ensure all abbreviations are defined at first mention.
3. The text is dense and would benefit from one or two more summary figures or tables. Though Table 1 provides a clear and informative summary of major biomarkers and their clinical utility, it already addresses part of the need for data visualization. To further enhance readability, consider adding a schematic figure that categorizes biomarkers by type and detection method, and maps them to their diagnostic or therapeutic applications, which would provide readers with an intuitive overview of the biomarker landscape. This can be achieved with a simple schematic figure and does not require extensive new data collection.
Author Response
Major Concerns
Comment 1: The review is informative but remains largely descriptive, focusing on listing biomarkers and their clinical implications without sufficient critical comparison of their relative performance and readiness for clinical use. Table 1 provides useful information on classical biomarkers but does not cover emerging biomarker classes (e.g., microRNAs, ctDNA, proteomics, metabolomics) or contrast their strengths, weaknesses, and validation status. Consider expanding the existing table or adding a comparative summary that includes sensitivity, specificity, clinical validation stage, advantages, and limitations of each biomarker class to enhance the analytical depth of the review.
Response 1: Thank you for this valuable suggestion. In response, we have substantially expanded the analytical section of the review and added three new tables that provide a comparative summary of classical
and emerging biomarker classes, including microRNAs, ctDNA, proteomic and metabolomic
markers. The tables now include information on their sensitivity, specificity, clinical validation
stage, advantages, and limitations, thereby enhancing the analytical depth and comparative
dimension of the manuscript.
Comment 2: The manuscript summarizes biomarker data effectively but provides little mechanistic explanation linking these biomarkers to ovarian cancer pathophysiology. For example, the section on microRNAs could explain how the miR-200 family affects EMT pathways or how lipid profile changes relate to tumor metabolism. My suggestion is to briefly integrate molecular pathways or mechanistic models that explain why these biomarkers change in ovarian cancer. Integrating even brief mechanistic insights would elevate the review from a descriptive summary to a more analytical and impactful contribution.
Response 2: Thank you very much for your insightful and constructive feedback. I have carefully revised the manuscript to address your suggestions. Specifically, I have incorporated mechanistic explanations linking the discussed biomarkers to ovarian cancer pathophysiology. The section on microRNAs now includes a brief discussion of how the miR-200 family regulates epithelial–mesenchymal transition (EMT) pathways, while the part concerning lipid metabolism has been expanded to highlight how alterations in lipid profiles contribute to tumor growth, metastasis, and chemoresistance. I believe these additions enhance the analytical depth of the review and strengthen its overall scientific impact.
Comment 3: The text acknowledges the heterogeneity of ovarian tumors but does not systematically discuss how different biomarker classes perform in specific histological subtypes (e.g., HGSOC vs. CCOC vs. MOC). Expand on subtype-specific biomarker utility — this is particularly important for clinical translation.
Response 3: We thank the reviewer for this insightful comment emphasizing the importance of illustrating biomarker performance across ovarian carcinoma subtypes. In response, we have added new summary tables presenting the diagnostic, prognostic, and predictive performance of different biomarker classes (serum, immunohistochemical, molecular, and emerging omics-based markers) within major histologic subtypes. These tables aim to improve clarity, facilitate comparison between tumor types, and enhance the translational relevance of the review.
Comment 4: Many statements rely on clusters of references, but the source quality, study type, and strength of evidence are not clear. Where possible, specify whether cited studies are meta-analyses, clinical trials, or basic research to strengthen the manuscript’s scientific rigor.
Response 4: The references have been reviewed and clarified where possible to enhance the manuscript’s scientific rigor.
Minor Concerns
Comment 1: Several sentences are too long or repetitive, which can reduce readability.
Example: “MicroRNAs present in exosomes are promising, non-invasive biomarkers that can be used both to monitor response to treatment and to assess the risk of cancer recurrence.” → Could be shortened for clarity. It is better to simplify and split long sentences.
Response 1: We appreciate your comment. Following the suggestion, we carefully re-read the entire text and identified several sentences that were indeed too long or complex. These sentences have now been simplified or divided to improve readability and enhance the overall flow of the manuscript. We are grateful for this remark, as it helped us make the manuscript clearer and more reader-friendly.
Comment 2: Terms like “ctDNA,” “HGSOC,” “MOC,” and “CCOC” are used without a consistent prior definition. Ensure all abbreviations are defined at first mention.
Response 2: We sincerely thank the Reviewer for this helpful remark. To improve the clarity and consistency of the manuscript, we have carefully reviewed all abbreviations used in the manuscript. All abbreviations including ctDNA, HGSOC, MOC, and CCOC. are properly listed in the "Abbreviations" section at the end of the manuscript, ensuring clarity and consistency for readers.
Comment 3: The text is dense and would benefit from one or two more summary figures or tables. Though Table 1 provides a clear and informative summary of major biomarkers and their clinical utility, it already addresses part of the need for data visualization. To further enhance readability, consider adding a schematic figure that categorizes biomarkers by type and detection method, and maps them to their diagnostic or therapeutic applications, which would provide readers with an intuitive overview of the biomarker landscape. This can be achieved with a simple schematic figure and does not require extensive new data collection.
Response 3: We appreciate this helpful suggestion. While we did not include an additional schematic figure
categorizing biomarkers by detection method, we have enhanced the visual presentation of the
manuscript by adding three new tables summarizing biomarker classes, their diagnostic and
prognostic performance, and validation status. Additionally, we included a new schematic figure
illustrating the MAPK pathway activation mechanism in ovarian cancer, which visually supports
the molecular context discussed in the text.
Reviewer 3 Report
Comments and Suggestions for Authors
The topic is highly relevant for clinicians and researchers in gynecologic oncology, and the authors have compiled a large volume of information in an orderly and structured way. The effort and dedication behind this review are evident. Nevertheless, the manuscript would benefit from a major revision to improve its scientific precision, clarity, and overall impact.
1. Structure and organization
The manuscript follows a logical progression—from general background and tumor classification to classical biomarkers, novel approaches, and future perspectives—but it is excessively long and at times repetitive. Some subsections (e.g., 2. Classification of Ovarian Tumors and Associated Biomarkers) include extensive epidemiologic and histologic detail that could be shortened. The review would be more effective if focused on the molecular and translational aspects of biomarker application rather than reiterating morphological descriptions or survival statistics already well established in the literature.
2. Depth of analysis
While the review is comprehensive, it remains largely descriptive. The value of the paper would increase considerably if the authors integrated a more critical and comparative approach. For example:
-
Discuss how classical markers (CA-125, HE4) compare in diagnostic algorithms such as ROMA or RMI, highlighting limitations in premenopausal vs. postmenopausal patients.
-
Critically evaluate the clinical readiness of emerging biomarkers (microRNAs, ctDNA, metabolomic panels) — which of them are validated in prospective studies, and which remain exploratory.
-
Consider including a table summarizing biomarker performance (sensitivity, specificity, AUC values, level of validation, and clinical availability).
3. Scientific rigor and referencing
The review cites a very high number of references, many of them recent and appropriate. However, several are descriptive or single-center studies, while fewer systematic reviews or meta-analyses are cited. It would strengthen the manuscript to replace or complement older or less robust citations with high-impact, peer-reviewed sources (e.g., Nat Rev Clin Oncol, JCO, Gynecol Oncol, Clin Cancer Res).
In several sections, reference ranges (e.g., “sensitivity 64-75%”) appear without explicit citation or numerical support in the reference list—these should be precisely referenced to maintain scientific traceability.
4. Novelty and added value
The review brings together valuable information but adds limited novelty. It would be useful for the authors to emphasize what differentiates this work from existing reviews on ovarian biomarkers—perhaps focusing on the correlation between tumor type and biomarker profile as the central narrative thread. A concise comparative discussion of epithelial vs. germ-cell vs. sex cord-stromal biomarkers could add originality.
5. Figures, tables, and synthesis tools
Currently, the only summarized table (Table 1) is useful but insufficient. Additional schematic figures could greatly enhance readability—for example:
-
A graphical summary of biomarker pathways by tumor subtype.
-
A flow diagram illustrating how classical and novel biomarkers integrate into current diagnostic algorithms.
-
A concise “clinical utility map” of markers validated for diagnosis, prognosis, or therapeutic monitoring.
6. English language and style
The English is generally understandable but would benefit from professional editing. The style is verbose, with long sentences and redundancy that affect flow and clarity. Streamlining the text and improving transitions between sections would make the manuscript more engaging and easier to read. MDPI’s English editing service or similar support is strongly recommended.
7. Conclusions and clinical perspective
The conclusions appropriately emphasize the potential of multi-omics integration and AI-based platforms but could be more concise and clinically oriented. It would help to end the review with clear, evidence-based take-home messages—what biomarkers are ready for routine use, which remain experimental, and what steps are needed for future clinical implementation.
8. Ethical and formal aspects
No ethical or methodological concerns are identified.
The English is generally understandable but would benefit from professional editing. The style is verbose, with long sentences and redundancy that affect flow and clarity. Streamlining the text and improving transitions between sections would make the manuscript more engaging and easier to read. MDPI’s English editing service or similar support is strongly recommended.
Author Response
Comment 1: Structure and organization
The manuscript follows a logical progression—from general background and tumor classification to classical biomarkers, novel approaches, and future perspectives—but it is excessively long and at times repetitive. Some subsections (e.g., 2. Classification of Ovarian Tumors and Associated Biomarkers) include extensive epidemiologic and histologic detail that could be shortened. The review would be more effective if focused on the molecular and translational aspects of biomarker application rather than reiterating morphological descriptions or survival statistics already well established in the literature.
Response 1: We appreciate the reviewer’s valuable feedback regarding the balance between histopathological and biomarker-related content. In the revised version, we have streamlined and condensed the histological descriptions to reduce redundancy and enhance focus on the biomarker-related aspects of each ovarian tumor subtype. This modification strengthens the manuscript’s alignment with its stated aim of emphasizing biomarker performance and clinical translation.
Comment 2: Depth of analysis
While the review is comprehensive, it remains largely descriptive. The value of the paper would increase considerably if the authors integrated a more critical and comparative approach. For example:
Discuss how classical markers (CA-125, HE4) compare in diagnostic algorithms such as ROMA or RMI, highlighting limitations in premenopausal vs. postmenopausal patients.
Response 2: Thank you for this comment. Information comparing classical markers (CA-125, HE4) within diagnostic algorithms such as ROMA and RMI, including their limitations in premenopausal and postmenopausal patients, is provided in Chapter 4 of the manuscript.
Comment 3: Critically evaluate the clinical readiness of emerging biomarkers (microRNAs, ctDNA, metabolomic panels) — which of them are validated in prospective studies, and which remain exploratory.
Response 3: Thank you for this valuable remark. The discussion on the clinical readiness and validation status of emerging biomarkers, including microRNAs, ctDNA, and metabolomic panels—distinguishing between those validated in prospective studies and those still considered exploratory—is included in Chapter 4 of the manuscript.
Comment 4: Consider including a table summarizing biomarker performance (sensitivity, specificity, AUC values, level of validation, and clinical availability).
Response 4: Thank you for this helpful suggestion. In response, we have added three new tables summarizing biomarker performance, including sensitivity, specificity, AUC values, level of validation, and clinical applicability. These tables provide a comprehensive overview of both classical and emerging biomarkers, thereby enhancing the analytical clarity of the manuscript.
Comment 5: Scientific rigor and referencing
The review cites a very high number of references, many of them recent and appropriate. However, several are descriptive or single-center studies, while fewer systematic reviews or meta-analyses are cited. It would strengthen the manuscript to replace or complement older or less robust citations with high-impact, peer-reviewed sources (e.g., Nat Rev Clin Oncol, JCO, Gynecol Oncol, Clin Cancer Res).
In several sections, reference ranges (e.g., “sensitivity 64-75%”) appear without explicit citation or numerical support in the reference list—these should be precisely referenced to maintain scientific traceability.
Response 5: References have been updated as suggested.
Comment 6: Novelty and added value
The review brings together valuable information but adds limited novelty. It would be useful for the authors to emphasize what differentiates this work from existing reviews on ovarian biomarkers—perhaps focusing on the correlation between tumor type and biomarker profile as the central narrative thread. A concise comparative discussion of epithelial vs. germ-cell vs. sex cord-stromal biomarkers could add originality.
Response 6: We thank the reviewer for this insightful comment. In response, we have added new summary tables presenting the diagnostic, prognostic, and predictive performance of different biomarker classes (serum, immunohistochemical, molecular, and emerging omics-based markers) within major histologic subtypes. These tables aim to improve clarity, facilitate comparison between tumor types, and enhance the translational relevance of the review.
Comment 7: Figures, tables, and synthesis tools
Currently, the only summarized table (Table 1) is useful but insufficient. Additional schematic figures could greatly enhance readability—for example:
A graphical summary of biomarker pathways by tumor subtype.
A flow diagram illustrating how classical and novel biomarkers integrate into current diagnostic algorithms.
A concise “clinical utility map” of markers validated for diagnosis, prognosis, or therapeutic monitoring.
Response 7: Thank you for this helpful suggestion. In response, an additional schematic figure illustrating the constitutive activation of the MAPK pathway and its downstream effects (Figure 1) has been included. This visual representation highlights key molecular events and their clinical implications, thereby enhancing the readability and conceptual understanding of the discussed biomarker pathways
Comment 8: English language and style
The English is generally understandable but would benefit from professional editing. The style is verbose, with long sentences and redundancy that affect flow and clarity. Streamlining the text and improving transitions between sections would make the manuscript more engaging and easier to read. MDPI’s English editing service or similar support is strongly recommended.
Response 8: Thank you for the comment.
Comment 9: Conclusions and clinical perspective
The conclusions appropriately emphasize the potential of multi-omics integration and AI-based platforms but could be more concise and clinically oriented. It would help to end the review with clear, evidence-based take-home messages—what biomarkers are ready for routine use, which remain experimental, and what steps are needed for future clinical implementation.
Response 9: Conclusions have been added.
Comment 10: Ethical and formal aspects
No ethical or methodological concerns are identified.
Response 10: Thank you for the comment, but these aspects do not fit into the scope of our review.
Reviewer 4 Report
Comments and Suggestions for Authors
The authors of the paper made a deep scan of the literature on the chosen topic and presented many new information. But in some places, the data are not well organized. To better communicate to the reader the importance of these markers in ovarian cancer diagnosis and follow-up, improvements are needed in all chapters of the manuscript.
A fragmentation of chapter 2 in subchapters could be benefic, and more synthetic tables like table 1 should be inserted, for each ovarian cancer type. A systematic resettlement of the marker-linked information is needed in Chapter 3 instead of a pure enumeration. In some paragraphs the clinical presentation of the ovarian cancer types is longer than the description of the markers.
The paper need an extensive revision. See below some examples, comments:
Page 2, row 31 the sentence: “The ovary is an anatomical region that predominates in the presence of both physiological and pathological transformations” needs reformulation
Page2, row 35: “frequently reported in 2022” what is the importance of the year 2022 in the context of benign ovarian tumors frequency?
Some references (eg.2) are inappropriate; authors should display data from broader databases, inclusive reviews published in highly cited landmark publications on gynecological diseases topic.
Same for reference 1 and 5, they refer to single region’s results, they are not general comprehensive data to be used for biomarker development.
Page 2, row 49: “Early-stage diagnosis is possible in only 20–25% of patients”- citation is needed here, and a clarification is necessary: this is a present situation, not an upper limit. Serious efforts are being made worldwide to improve it.
Page 3, rows 54-57: “Determining the nature of the lesion—benign or malignant—is critical [13]. Diagnostic differentiation involves analysis of macroscopic, microscopic, and immunohistochemical properties and can often be challenging. Dundr et al. describe difficulties in distinguishing primary mucinous ovarian tumors from metastases due to overlapping characteristics [14].”
To distinguish between primary tumors and metastases is a totally different approach than discriminating between benign and cancer lesions. These two sentences do not match. If authors want to discuss about markers for metastases, they should write a separate paragraph about.
Page 3, rows 58-70: authors mixed serum markers with histological markers in the same paragraph; a clear demarcation should be made for these markers
Page 3, row 79: “and regional disparities” – please explain, are they protocols with regional restrictions? If so, a citation is needed. As well, to develop a new protocol which contains regional insights will have difficulty obtaining approvals
Page 4: in the whole chapter 2, it is important to establish which markers are measured from plasma/serum/ascites or tissues resulted from biopsies of tissue resections
Page 5: a short description of IMP3, Napsin A, HNF-1β, IFITM1, MUC5AC … and other less known markers is necessary (mRNA binding protein, aspartic protease, …); their role in ovarian cancer and other general considerations.
Page 6, row 217: “Key molecular biomarkers in LGSC are MAPK pathway mutations.” Later, the authors mention about KRAS, BRAF and NRAS- they are genes which control the pathway, the genes are mutated, not the pathway.
Page 6- in LGSC, teratomas, and other cancer types which are predictive and which are prognostic markers among the enumerated many markers?
Page7, row 288- SALL4 is expressed in the cells together with the other mentioned markers or separately?
Page 8, row 336: β-hCG abbreviation should be explained here; row 358 the itemized, bulleted listing is not appropriate here
Page 9, row 373- “SF-1, inhibin-α, WT1, calretinin”- explain these markers. The authors should assert more on the markers, which are very briefly described in compare with the clinical data presented;
Pages 12-14: in chapter3 information about microARN, metabolites, lipids, cDNA and other categories must be clearly delimited.
Chapter 3 has no paragraphs. The authors collected a large amount of data, they should be capitalized and made more attractive for the reader.
Page 13, row 491: “PPA1, BCAT1 and MCM4, while decreased levels are observed in MUC5B, SLC4A1 and tenascin-XB (TNXB), among others. (…) proteins, such as WFDC2 (also known as HE4), KRT19 and RBFOX3”
Page 13,Row 530 “ NOTCH1, 530NOTCH2, NOTCH3, NOTCH4, DLL1, JAG2, HES1) and Wnt (CTNNB1)”
All the above molecules should be described briefly, the abbreviation explained. Are they blood markers, tumor epitopes, mutated proteins? They are down-or upregulated?
Page14, row 562 ”New biomarkers such as microRNA, ctDNA, metabolites, proteomics (e.g. LPC, glycoproteins) and gene expression and transcriptomics have the potential to profoundly personalise treatment” Sentences such the above have no connection with the title of the chapter.
No clinical trial was mentioned, although they are trials which try to employ new biomarkers.
Pages 15-16, Chapter 5 - specific data should be included on which biomarker or which biomarker family has a real potential to be applied.
Author Response
Comment 1: A fragmentation of chapter 2 in subchapters could be benefic, and more synthetic tables like table 1 should be inserted, for each ovarian cancer type. A systematic resettlement of the marker-linked information is needed in Chapter 3 instead of a pure enumeration. In some paragraphs the clinical presentation of the ovarian cancer types is longer than the description of the markers.
Response 1: Revisions have been made accordingly.
Comment 2: Page 2, row 31 the sentence: “The ovary is an anatomical region that predominates in the presence of both physiological and pathological transformations” needs reformulation
Response 2: Thank you for your suggestion. We have revised the sentence to “The ovary is an organ where physiological and pathological changes frequently occur”.
Comment 3: Page2, row 35: “frequently reported in 2022” what is the importance of the year 2022 in the context of benign ovarian tumors frequency?
Response 3: Thank you for your comment. We have removed the sentence due to limited clinical relevance and to improve focus on more significant aspects of the topic.
Comment 4: Some references (eg.2) are inappropriate; authors should display data from broader databases, inclusive reviews published in highly cited landmark publications on gynecological diseases topic. Same for reference 1 and 5, they refer to single region’s results, they are not general comprehensive data to be used for biomarker development.
Response 4: Thank you for your careful observation. Reference 1 did not include any epidemiological data, as it was cited solely to support an introductory statement rather than to present specific data. Reference 2
has been removed, and global data from the comprehensive GLOBOCAN database have been
incorporated to ensure broader and more representative coverage of the topic.
Comment 5: Page 2, row 49: “Early-stage diagnosis is possible in only 20–25% of patients”- citation is needed here, and a clarification is necessary: this is a present situation, not an upper limit. Serious efforts are being made worldwide to improve it.
Response 5: Thank you for this insightful remark. The sentence has been revised for clarity and accuracy to reflect the current situation rather than an upper limit. A relevant citation has also been added to support this
statement.
Comment 6: Page 3, rows 54-57: “Determining the nature of the lesion—benign or malignant—is critical [13]. Diagnostic differentiation involves analysis of macroscopic, microscopic, and immunohistochemical properties and can often be challenging. Dundr et al. describe difficulties in distinguishing primary mucinous ovarian tumors from metastases due to overlapping characteristics [14].”
To distinguish between primary tumors and metastases is a totally different approach than discriminating between benign and cancer lesions. These two sentences do not match. If authors want to discuss about markers for metastases, they should write a separate paragraph about.
Response 6: Thank you for your comment. We have revised the paragraph to maintain focus on the distinction between benign and malignant lesions. The reference to metastases has been removed to avoid shifting attention from the primary diagnostic challenge. The text now reads: Determining the nature of the lesion—benign or malignant—is critical. Diagnostic differentiation involves analysis of macroscopic, microscopic, and immunohistochemical properties and can often be challenging.
Comment 7: Page 3, rows 58-70: authors mixed serum markers with histological markers in the same paragraph; a clear demarcation should be made for these markers.
Response 7: The section has been revised as suggested.
Comment 8: Page 3, row 79: “and regional disparities” – please explain, are they protocols with regional restrictions? If so, a citation is needed. As well, to develop a new protocol which contains regional insights will have difficulty obtaining approvals
Response 8: A new citation has been added, and the section has been revised accordingly.
Comment 9: Page 4: in the whole chapter 2, it is important to establish which markers are measured from plasma/serum/ascites or tissues resulted from biopsies of tissue resections
Response 9: The data have been presented in new tables.
Comment 10: Page 5: a short description of IMP3, Napsin A, HNF-1β, IFITM1, MUC5AC … and other less known markers is necessary (mRNA binding protein, aspartic protease, …); their role in ovarian cancer and other general considerations.
Response 10: Thank you very much for your valuable and detailed feedback. I have carefully revised the section and added concise descriptions of IMP3, Napsin A, HNF-1β, IFITM1, MUC5AC including their molecular characteristics (such as mRNA-binding function or aspartic protease activity) and their relevance in ovarian cancer. These additions provide a clearer understanding of their biological roles and clinical significance. I sincerely appreciate your insightful comments, which have greatly contributed to enhancing the scientific depth and completeness of the manuscript.
Comment 11: Page 6, row 217: “Key molecular biomarkers in LGSC are MAPK pathway mutations.” Later, the authors mention about KRAS, BRAF and NRAS- they are genes which control the pathway, the genes are mutated, not the pathway.
Response 11: Thank you very much for your precise and insightful comment. I have carefully revised the sentence to accurately reflect the molecular mechanism described. The text now specifies that mutations occur in the KRAS, BRAF, and NRAS genes, which are key regulators of the MAPK signaling pathway, rather than in the pathway itself. I greatly appreciate your valuable feedback, which has helped to improve the scientific accuracy and clarity of the manuscript.
Comment 12: Page 6- in LGSC, teratomas, and other cancer types which are predictive and which are prognostic markers among the enumerated many markers?
Response 12: Thank you very much for your valuable comment. In response, I have created a table to systematically organize the marker-related information. The table categorizes the markers according to cancer subtype and indicates whether each marker has predictive or prognostic value.
Comment 13: Page7, row 288- SALL4 is expressed in the cells together with the other mentioned markers or separately?
Response 13: Thank you for your comment. I have revised the manuscript according to the suggestion.
Comment 14: Page 8, row 336: β-hCG abbreviation should be explained here; row 358 the itemized, bulleted listing is not appropriate here
Response 14: You’re absolutely right, the explanation of the abbreviation was lacking and was respectively explicated
Comment 15: Page 9, row 373- “SF-1, inhibin-α, WT1, calretinin”- explain these markers. The authors should assert more on the markers, which are very briefly described in compare with the clinical data presented
Response 15: The markers have been further explained as suggested.
Comment 16: Pages 12-14: in chapter3 information about microARN, metabolites, lipids, cDNA and other categories must be clearly delimited.
Response 16: Thank you for your suggestion, the categories have been clearly delimited.
Comment 17: Chapter 3 has no paragraphs. The authors collected a large amount of data, they should be capitalized and made more attractive for the reader.
Response 17: Thank you for your valuable feedback.I have revised Chapter 3 accordingly - it has now been divided into coherent paragraphs, and the collected data have been properly formatted, emphasized, and presented in a more engaging and reader-friendly manner to enhance the overall clarity and academic quality of the chapter.
Comment 18: Page 13, row 491: “PPA1, BCAT1 and MCM4, while decreased levels are observed in MUC5B, SLC4A1 and tenascin-XB (TNXB), among others. (…) proteins, such as WFDC2 (also known as HE4), KRT19 and RBFOX3”
Response 18: Done, the protein descriptions have been clarified as suggested.
Comment 19: Page 13,Row 530 “ NOTCH1, 530NOTCH2, NOTCH3, NOTCH4, DLL1, JAG2, HES1) and Wnt (CTNNB1)”
Response 19: The gene markers have been clarified as suggested.
Comment 20: All the above molecules should be described briefly, the abbreviation explained. Are they blood markers, tumor epitopes, mutated proteins? They are down-or upregulated?
Response 20: The above molecules have been clarified as suggested.
Comment 21: Page14, row 562 ”New biomarkers such as microRNA, ctDNA, metabolites, proteomics (e.g. LPC, glycoproteins) and gene expression and transcriptomics have the potential to profoundly personalise treatment” Sentences such the above have no connection with the title of the chapter.
Response 21: Thank you very much for your thoughtful and constructive feedback. I have carefully reviewed the section and deleted the indicated sentence to ensure that the content remains consistent with the chapter title and maintains a clear thematic focus. I truly appreciate your valuable remarks, which have contributed to improving the overall quality and coherence of the manuscript.
Comment 22: No clinical trial was mentioned, although they are trials which try to employ new biomarkers.
Response 22: Thank you very much for your insightful comment. In response, I have added information about a clinical trial related to the use of ctDNA. While no clinical trials were previously mentioned, I agree that it is important to highlight ongoing efforts to employ new biomarkers in clinical settings. Including this trial helps to illustrate the translational potential of ctDNA and reflects the current direction of research in this area.
Comment 23: Pages 15-16, Chapter 5 - specific data should be included on which biomarker or which biomarker family has a real potential to be applied.
Response 23: Thank you very much for your valuable comment. The specific information regarding biomarkers and biomarker families with real potential for application has been included in Chapter 4.
Round 2
Reviewer 2 Report
Comments and Suggestions for Authors
I think the schematic figure illustrating the MAPK pathway is too simple; everyone is already familiar with the molecules in this cascade. The authors are encouraged to add more detailed information, such as key molecular mutations associated with LGSC, to make the figure more informative and specific to their study.
The other revisions appear generally acceptable, though I suggest the authors carefully review all tables and data to ensure accuracy and consistency.
Author Response
Comment 1: I think the schematic figure illustrating the MAPK pathway is too simple; everyone is already familiar with the molecules in this cascade. The authors are encouraged to add more detailed information, such as key molecular mutations associated with LGSC, to make the figure more informative and specific to their study.
Response 1: We sincerely thank the reviewer for this valuable and insightful comment. In response to the suggestion, we have revised the schematic figure of the MAPK pathway to include additional details, specifically highlighting key molecular mutations associated with LGSC. We believe that these modifications make the figure more informative and directly relevant to our work.
Comment 2: The other revisions appear generally acceptable, though I suggest the authors carefully review all tables and data to ensure accuracy and consistency.
Response 2: We sincerely thank the reviewer for this comment.
Reviewer 3 Report
Comments and Suggestions for Authors
I would like to thank the authors for the careful and constructive revision of the manuscript. The improvements are clear and well aligned with the previous review recommendations. The structure is now more focused, the redundancy in some descriptive sections has been reduced, and the incorporation of comparative tables significantly increases the clarity and clinical applicability of the review. The expanded discussion of circulating biomarkers, proteomic/metabolomic signatures, and microRNA-based approaches is now better integrated and provides a more cohesive narrative of classical and emerging biomarker strategies.
The manuscript is scientifically sound, well referenced, and now offers a more balanced interpretation of the evidence. The clinical relevance is clearer, particularly in sections where biomarker performance is contextualized by tumor subtype and intended clinical use (diagnostic, prognostic, or predictive).
Minor points to address before final acceptance:
-
Conclusion:
Consider shortening the final section to highlight 3–5 clear take-home messages, especially regarding which biomarkers are ready for clinical application vs. those that remain investigational. -
Nomenclature consistency:
Standardize formatting for certain markers (e.g., HNF-1β / HNF1B, miRNA notation, protein name capitalization) to ensure uniformity across the manuscript. -
Language polishing:
The English has improved substantially; however, a final light editing pass would help streamline some long sentences and improve overall fluency.
Overall, the revised version represents a clear improvement and provides a useful and well-structured contribution to the field.
Comments on the Quality of English LanguageThe English has clearly improved; however, a final, targeted language edit will further enhance clarity, consistency, and scientific style. Below are specific, actionable points:
-
Sentence length & concision
Several sentences remain long or layered with subordinate clauses. Prefer two shorter sentences to one very long sentence.
• Example: “As far as immunohistochemical markers are concerned, nearly all HGSOC cases show abnormal p53 expression due to TP53 mutations.” → “Nearly all HGSOC show abnormal p53 immunostaining, reflecting underlying TP53 mutations.” -
Terminology & notation consistency
Keep one form throughout.
• HNF-1β vs HNF1β → choose HNF-1β across text, tables and figures.
• miRNA notation: use miR-200c, miR-371a-3p, let-7g (non-italic), with a non-breaking hyphen (-) and consistent subscript formatting in tables.
• Gene vs protein: genes in italics (e.g., TP53, KRAS, BRCA1/2), proteins non-italic (p53, KRAS).
• β-hCG: use Greek beta (β) consistently; avoid mixing β-hCG / beta-hCG / β HCG.
• ctDNA/“circulating tumor DNA”: define at first mention and use the abbreviation thereafter. Same for OGCT, SCST, HGSOC, LGSC, etc.
• “Multi-marker” vs “multimarker” → choose one (journal style typically “multi-marker”).
• “meta-analyses” (not “metanalyses”). -
Numerical style & symbols
• Replace “~” with “approximately” in narrative text.
• Use en dashes for ranges (e.g., 63–83%) and ≥ / ≤ instead of “>= / <=”.
• Use the decimal point (English style) and avoid mixing comma/point formats.
• Spell out numbers at the start of sentences (“Five-year survival…”). -
Units, abbreviations & first-use definitions
• At first mention of a marker or metric, provide unit and cutoff (e.g., “CA-125, cut-off 35 U/mL”).
• When reporting accuracy (“AUC 0.96”), consider “AUC = 0.96” and, where available, add 95% CI.
• Ensure IU/mL spacing and capitalization are consistent. -
Hedging, claims & precision
• Prefer cautious verbs (“is associated with”, “suggests”, “may improve”) over absolute claims when evidence is observational.
• Where you report performance (“sensitivity 64–75%”), cite precisely the source(s) immediately after the sentence. -
Parallelism & list hygiene
• Ensure lists use parallel grammatical structure and consistent punctuation.
• Example: “diagnostic, prognostic, and for therapeutic monitoring” → “diagnostic, prognostic, and monitoring”. -
Punctuation & style
• Adopt one variety (American/British) and keep it throughout; the manuscript currently mixes both.
• Use the serial (Oxford) comma consistently or omit it consistently—current usage is mixed.
• Replace fillers (“As far as…”, “What is more,”, “Last but not least,”) with neutral scientific connectors (“Additionally,”, “Moreover,”, “Finally,”). -
Section cohesion & transitions
• Improve topic sentences at the start of subsections to signal scope (“This section summarizes circulating biomarkers by histotype and clinical use.”).
• Add brief transition lines when switching from epithelial to germ-cell or sex cord-stromal tumors. -
Examples of micro-edits
• “High p53 expression (>=70% positive nuclei or complete absence is typical in HGSOC…)” → “Abnormal p53 immunostaining—either diffuse strong nuclear positivity (≥70% nuclei) or complete absence—is typical of HGSOC…”
• “Most metanalyses studies reporting between 14% and 19%.” → “Most meta-analyses report 14–19%.”
• “As far as the time of the paper…” → “At the time of writing…”
• “and as well as” (redundant) → use “and” or “as well as”, not both.
• “Figure 1. Constitutive MAPK pathway activation.” → Make sure the legend is complete, self-contained, and matches the in-text call-out (define abbreviations used in the figure). -
Consistency in tumor names
• Use one form: high-grade serous ovarian carcinoma (HGSOC), endometrioid ovarian carcinoma (EOC), clear cell ovarian carcinoma (CCOC), mucinous ovarian carcinoma (MOC), low-grade serous carcinoma (LGSC)—and then abbreviations.
Overall, the manuscript is now clear and readable; implementing the style and micro-consistency fixes above will give it a fully polished finish suitable for publication.
Author Response
We would like to sincerely thank the Reviewer for the positive and encouraging evaluation of our revised manuscript. We truly appreciate the acknowledgment of the improved structure, reduced redundancy, and inclusion of comparative tables that enhance clarity and clinical applicability.
We are also grateful for the Reviewer’s recognition of the expanded and better-integrated discussion on circulating biomarkers, proteomic/metabolomic signatures, and microRNA-based approaches.
We are pleased that the manuscript is now considered scientifically sound, well referenced, and more clinically relevant. We greatly value the Reviewer’s constructive feedback and kind remarks, which have been instrumental in improving the quality of our work.
Comment 1: Consider shortening the final section to highlight 3–5 clear take-home messages, especially regarding which biomarkers are ready for clinical application vs. those that remain investigational.
Response 1: We thank the Reviewer for this valuable suggestion. The final section emphasizes 3 key take-home messages. These clearly distinguish biomarkers that are ready for clinical implementation from those that remain under investigation. We appreciate this helpful recommendation, which has improved the focus and clarity of the conclusion.
Comment 2: Standardize formatting for certain markers (e.g., HNF-1β / HNF1B, miRNA notation, protein name capitalization) to ensure uniformity across the manuscript.
Response 2: We thank the Reviewer for this helpful observation. The formatting of all biomarkers has been standardized throughout the manuscript, including consistent notation for HNF-1β/HNF1B, microRNAs, and protein names.
Comment 3: The English has improved substantially; however, a final light editing pass would help streamline some long sentences and improve overall fluency.
Response 3: We sincerely thank the Reviewer for the detailed and constructive language recommendations. A thorough final editing pass has been performed to address all points, including sentence concision, terminology and notation consistency, numerical and unit formatting, as well as style and punctuation uniformity. We have carefully implemented these improvements to enhance clarity, fluency, and overall readability of the manuscript.
Reviewer 4 Report
Comments and Suggestions for Authors
The authors made several changes in the manuscript, the revised paper contains more information, better structured as in the previous version. Tables, references and improved graphs were added, the list of abbreviations was improved. The authors answered to all my comments, one by one, and the edited paragraphs are marked with red color. Therefore I suggest publishing the manuscript, this revised form is more attractive to the readers.
Author Response
Comment 1: The authors made several changes in the manuscript, the revised paper contains more information, better structured as in the previous version. Tables, references and improved graphs were added, the list of abbreviations was improved. The authors answered to all my comments, one by one, and the edited paragraphs are marked with red color. Therefore I suggest publishing the manuscript, this revised form is more attractive to the readers.
Response 1: We sincerely thank the reviewer for their positive evaluation and kind recommendation for publication.